# A New View of Heat Wave Dynamics and Predictability over the Eastern Mediterranean

Assaf Hochman[1], Sebastian Scher[2], Julian Quinting[1], Joaquim G. Pinto[1], Gabriele Messori[2, 3]

[1] Department of Tropospheric Research, Institute of Meteorology and Climate Research, Karlsruhe Institute of Technology, Karlsruhe, Germany.
[2] Department of Meteorology and Bolin Centre for Climate Research, Stockholm University, Stockholm, Sweden.
[3] Department of Earth Sciences and Centre of Natural Hazards and Disaster Science (CNDS), Uppsala University, Uppsala, Sweden.

*Correspondence to*: Assaf Hochman (assaf.hochman@kit.edu)

**Abstract.** Skillful forecasts of extreme weather events have a major socio-economic relevance. Here, we compare two complementary approaches to diagnose the predictability of extreme weather: recent developments in dynamical systems theory and numerical ensemble weather forecasts. The former allows us to define atmospheric configurations in terms of their persistence and local dimension, which inform on how the atmosphere evolves to and from a given state of interest. These metrics may be used as proxies for the intrinsic predictability of the atmosphere, which only depends on the atmosphere's properties. Ensemble weather forecasts inform on the practical predictability of the atmosphere, which partly depends on the performance of the numerical model used. We focus on heat waves affecting the Eastern Mediterranean. These are identified using the Climatic Stress Index (CSI), which was explicitly developed for the summer weather conditions in this region and differentiates between heat waves (upper decile) and cool days (lower decile). Significant differences are found between the two groups from both the dynamical systems and the numerical weather prediction perspectives. Specifically, heat waves show relatively stable flow characteristics (high intrinsic predictability), but comparatively low practical predictability (large model spread/error). For 500 hPa geopotential height fields, the *intrinsic* predictability of heat waves is lowest at the event's onset and decay. We relate these results to the physical processes governing Eastern Mediterranean summer heat waves: adiabatic descent of the air parcels over the region and the geographical origin of the air parcels over land prior to the onset of a heat wave. A detailed analysis of the mid-August 2010 record-breaking heat wave provides further insights into the range of different regional atmospheric configurations conducive to heat waves. We conclude that the dynamical systems approach can be a useful complement to conventional numerical forecasts for understanding the dynamics and predictability of Eastern Mediterranean heat waves.

## 1. Introduction

Heat waves are recognized as a major natural hazard (e.g., Easterling *et al*., 2000), causing detrimental socio-economic impacts (e.g., Feeling the heat, 2018) including excess mortality (e.g., Batisti and Naylor, 2009; Benett *et al*., 2014; Peterson *et al*., 2013; Ballester *et al*., 2019), agricultural loss (e.g., Deryng *et al*., 2014) and ecosystem impairment (e.g., Williams, 2014; Caldeira *et al*., 2015). Moreover, heat waves are projected to increase in frequency, intensity and persistence under global warming (e.g., Meehl and Tebaldi, 2004; Stott *et al.,* 2004; Fischer and Schär, 2010; Seneviratne *et al*., 2012; Russo *et al.,* 2014). The Eastern Mediterranean has experienced several extreme heat waves in recent decades (e.g., Kuglitsch *et al*., 2010) and their frequency and intensity are expected to increase in the coming decades (e.g., Giorgi 2006; Seneviratne *et al*., 2012; Lelieveld *et al*., 2016; Hochman *et al*., 2018a) upon a background of regional warming and drying (e.g., Barchikovska *et al*., 2020).

The Eastern Mediterranean climate is characterized by wet conditions and mild air temperatures during the winter season and dry and hot weather conditions during summer (e.g., Goldreich *et al*., 2003; Kushnir *et al*., 2017). The summer season is characterized by very small inter-daily variability, which is attributable to the dominant and persistent influence of the Persian Trough and sub-tropical high-pressure systems. The interaction between these systems leads to persistent north – westerly winds of continental origin blowing across the Aegean Sea. These winds are known since ancient times as 'Etesian winds' (Tyrlis and Lelieveld, 2013). Together with the Mediterranean Sea breeze, moist air can be transported inland (Alpert *et al*., 1990; Bitan and Saaroni, 1992) as far as the Dead Sea (Kunin *et al*., 2019). On the upper levels of the Troposphere, large-scale subsidence is dominant, thus further hampering the development of clouds and precipitation (Rodwell and Hoskins, 1996; Ziv *et al*., 2004). In spite of this generally low variability, heat waves are not infrequent during the summer (Harpaz *et al*., 2014). Still, episodes when the temperature drops to below-normal values do occur, some of which are accompanied by summer rains (Saaroni and Ziv, 2000).

Saaroni *et al*. (2017) have detected weaknesses in the ability of earlier synoptic classifications (Alpert *et al*., 2004a; Dayan *et al*., 2012) to describe local weather conditions during the Eastern Mediterranean summer season. The authors proposed a 'Climatic Stress Index' (CSI), which is a combination of the national heat stress index, used operationally by the Israeli Meteorological Service, and the height of the marine inversion base height (see Sect. 2.2). The authors argued that this novel index improves the classification of heat wave days relative to earlier classifications and additionally links directly to the potential impacts.

A notable heat wave in recent years was the 2010 so-called "Russian heat wave", which caused ~55,000 excess deaths in Eastern Europe and Western Russia (e.g., Barriopedro *et al*., 2011; Katsafados *et al*., 2014). The 2010 Northern Hemisphere summer saw a strong and persistent blocking ridge at 500 hPa over the Middle East and Eastern Europe (e.g., Grumm 2011; Schneidereit *et al*., 2012; Quandt *et al*., 2017), leading to unprecedented temperatures at numerous locations (Barriopedro *et al*., 2011). The Eastern Mediterranean and Israel experienced a record-breaking (in temperature) heat wave during mid-August

of that year (https://ims.gov.il/sites/default/files/aug10.pdf), which interestingly coincided with what is considered the decay phase of the Russian heat wave (Quandt *et al*., 2019). In fact, the Zefat Har-Knaan station (Tab. S1; Fig. S1) recorded a temperature of 40.6°C; the highest temperature since 1939, while the Jerusalem station (Tab. S1; Fig. S1) logged a remarkable 41°C, the absolute record for this station since 1942. The ability to predict and issue appropriate warnings for these types of events, and more generally weather events lying in the tails of the respective distributions, is of crucial importance for mitigation of impacts on human life, agriculture and ecosystems (IPCC 2012; Siebert and Evert, 2014; Williams 2014).

A general framework that allows a quantitative understanding of processes leading to extreme temperatures during heat waves is that based on Lagrangian backward trajectories. In this framework, the temperature of an air parcel increases by: (i) adiabatic warming related to descent and (ii) diabatic heating including latent and sensible heat fluxes, short-wave, and long-wave radiation (Holton 2004). Recent studies revealed that extreme temperatures during heat waves are most often a combination of adiabatic warming related to descent and diabatic heating near the surface (e.g., Black *et al*., 2004; Bieli *et al*., 2015; Santos *et al*., 2015; Quinting and Reeder, 2017; Zschenderlein *et al*., 2019). The adiabatic warming is typically associated with upper-level ridges, which promote subsidence. The strongest diabatically-driven heating does not necessarily occur at the location of the heat wave itself, but rather in geographically remote regions (e.g., Quinting and Reeder, 2017; Quinting *et al*., 2018; Zschenderlein *et al*., 2019).

Focusing more directly on the prediction of the evolution of specific atmospheric configurations, which may lead to heat waves, one may consider a partly model-dependent perspective (*practical* predictability) or a model-independent perspective (*intrinsic* predictability; Melhauser and Zhang, 2012). The practical predictability is heavily reliant on the availability of initialization data (Lorenz 1963) and on the correct representation of relevant physical processes in the numerical model being used. However, it also reflects some characteristics of the atmospheric dynamics (e.g., Ferranti *et al*., 2015; Matsueda and Palmer, 2018). An often-used method for quantifying the practical predictability is the spread or skill of ensemble forecasts (e.g., Loken *et al*., 2019).

As opposed to the practical predictability, the intrinsic predictability only depends on the characteristics of the atmosphere itself. However, it is important to note that the atmosphere is influenced and sometimes even controlled by interactions with the land and oceans, albeit mostly at longer time scales than those considered in this study (Entin *et al*., 2000; Koster *et al*., 2010; Dirmeyer *et al*., 2018). Recent developments in dynamical systems theory allow us to quantify the intrinsic predictability of instantaneous atmospheric states using two metrics: persistence ($\theta^{-1}$) and local dimension ($d$). These reflect how the atmosphere evolves in the neighborhood of a state of interest (Faranda *et al*., 2017a). High (low) $\theta^{-1}$ ($d$) imply high intrinsic predictability, whereas low (high) $\theta^{-1}$ ($d$) suggest low intrinsic predictability. The two forms of atmospheric predictability depend on different factors, and therefore offer different information. While there is some relation between the two (e.g., Scher and Messori, 2018), one should not expect them to always match for individual cases (Hochman *et al*., 2020a).

In the present study, we perform a systematic dynamical systems investigation of the temporal evolution of Eastern Mediterranean summer heat waves, and evaluate whether this may provide insights complementary to a more conventional analysis of the numerical weather forecasts of such events. Specifically, we hypothesize that the dynamical systems analysis captures relevant features of these extremes, such as their persistence, which are not always reflected in the numerical weather forecast. The dynamical systems framework has recently been leveraged for the study of cold spell dynamics (Hochman *et al*., 2020a).

The paper is organized as follows: Sect. 2 provides a brief description of the methodology, including the used datasets, the CSI index, the dynamical systems and forecast skill metrics and the method for backtracking air parcels. Sect. 3 describes the dynamics of heat waves from both the dynamical system and the numerical weather prediction perspectives and further provides a detailed analysis of the mid-August 2010 heat wave over the Eastern Mediterranean as a case study. Finally, Sect. 4 provides the main conclusions and discusses ideas for future research.

## 2.  Data and methods

### 2.1 Data

The bulk of our analysis is based on the National Centers for Environmental Prediction/National Center for Atmospheric Research Reanalysis Project (NCEP/NCAR) daily and 6-hourly reanalysis data for 1979 – 2015 (satellite era), on a 2.5° × 2.5° horizontal grid (Kalnay *et al*., 1996). Faranda *et al*. (2017a) have shown that the conclusions one may infer from the dynamical systems analysis are generally insensitive to the dataset's horizontal spatial resolution, as long as the major structures characterizing the atmospheric field of interest are resolved. On the contrary, the air parcel tracking (Sect. 2.5) requires data on a relatively high horizontal and vertical grid-spacing. Air parcel trajectories are thus computed from 6-hourly ERA-Interim data for 1979 – 2015, on a 1° × 1° horizontal grid and 60 vertical levels (Dee *et al*., 2011).

The numerical forecasts are acquired from the Global Ensemble Forecast System (GEFS) reforecast v.2 dataset produced by NCEP/NCAR (Hamill *et al*., 2013). Operational Numerical Weather Prediction (NWP) models are frequently updated. Therefore, archives of operational NWP models are usually inhomogeneous, and thus are not appropriate for studying predictability over long time periods. This problem can be mitigated by using so-called reforecasts. For reforecasts, one fixed version of an NWP model is used in order to create a standardized set of past forecasts. The GEFS reforecast dataset provides a set of daily reforecasts from December 1984 to present. Each reforecast consists of a control forecast and a ten-member ensemble on a 0.5° × 0.5° grid spacing.

Finally, we make use of a homogenized station dataset over Israel to assess the forecasts. Instrumental meteorological records may be influenced by non-meteorological events, such as station relocation, defects in the instrumentation, environmental changes near the station etc. The detrimental effects these may have on the quality of the recorded data can be reduced by

homogeneity procedures (Aguilar *et al*., 2003). Our dataset includes five representative, homogenized stations in Israel with
130 an uninterrupted record of maximum temperatures over 1979 – 2015 (Tab. S1, Fig. S1; Yosef *et al*., 2018).

## 2.2 Heat wave definition according to the Climatic Stress Index (CSI)

Saaroni *et al*. (2017) have proposed a new index for classifying the summer days over the Eastern Mediterranean based on the 'environment to climate' approach (Yarnal 1993; Yarnal *et al*., 2001). The CSI is comprised of the national heat stress index,
used operationally by the Israel Meteorological Service, and the marine inversion base height, which is a major factor influencing the summer weather conditions over the Eastern Mediterranean (Ziv *et al*., 2004). The index suits the identification of heat waves as it is does not merely consider the daily temperature, but rather additional variables, e.g., humidity and circulation, which directly relate to the impacts of a heat wave on for example, human physiology (Epstein and Moran, 2006). Saaroni *et al*. (2017) have rigorously evaluated the CSI index with respect to observations and tested a variety of different
combinations of predictors, which ultimately resulted in a simple multiple regression equation:

$$CSI = 92.78 + 0.638T_{1000-850} - 0.178\Delta p - 1.08p_{Iraq}$$

Here, $T_{1000-850}$ is the average regional lower-level temperature over [31°N-34°N; 33°E-37°E]. $\Delta p$ is the average sea level pressure over [36°N-44°N; 42°E-54°E] subtracted from the average sea level pressure over [24°N-29°N; 33°E-37°E], which is an estimate for the intensity of the Etesian winds (see Sect. 1). $p_{Iraq}$ represents the average sea level pressure over northern
Iraq [35°N-44°N; 46°E-54°E], which is a proxy for the depth of the Persian Trough.

The analysis described in the next sections is specifically implemented for extremes of the CSI index, i.e., days during which the CSI exceeds the 90[th] percentile of the July and August climatological distribution (hereafter: 'upper 10% of CSI' or heat waves) versus days when the CSI is below the 10[th] percentile of the July and August distribution (hereafter 'lower 10% of CSI' or cool days). The onset of a heat wave (cool days) is taken to be the first day in which the CSI exceeds (subceeds) the 90[th]
(10[th]) percentile threshold at 12UTC (0 h time in the Figures), which ought to roughly match the time of maximum daily temperature. Alpert *et al.* (2004b) have argued that July and August represent the mid-summer months, in which the Persian Trough occurs on more than nine out of eleven days on average. For additional details on the computation of the CSI index and its evaluation, the reader is referred to Saaroni *et al.* (2017).

## 2.3 Dynamical systems metrics

We view the atmosphere as a chaotic dynamical system, and leverage a recently-developed method combining extreme value theory with Poincaré recurrences (Lucarini *et al*., 2016; Faranda *et al*., 2017a) to estimate the dynamical properties of atmospheric states. The temporal evolution of the atmosphere can be represented as a long trajectory in a suitably defined

phase space. When we use temporally discretised data, such as reanalysis data, we are effectively sampling this trajectory for a given time period, for example 6 or 24 hours. An example would be analysing daily latitude-longitude maps of Sea-Level Pressure (SLP) over the Eastern Mediterranean (technically a special Poincaré section of the full dynamics): each 2-D map corresponds to a single point along the aforementioned trajectory, for which we seek to compute instantaneous (in time) and local (in phase space) properties. We specifically consider two metrics, which describe instantaneous atmospheric states: the local dimension $d$ and the persistence $\theta^{-1}$. In order to compute the local dimension and persistence for a given state of interest $\xi$, which in our example would correspond to a specific SLP map in our dataset, we first define logarithmic returns as:

$$g(x(t),\xi) = -log\big(dist(x(t),\xi)\big)$$

where *dist* is the Euclidean distance between two vectors. Thus, we define $g$ such that it is large when the system is in states close to $\xi$, and small when the system is in states far from $\xi$. In other words, $g$ is large whenever the SLP map on a given day resembles the SLP map of the day corresponding to the state of interest $\xi$.

We next consider all cases in which $g$ exceeds a high threshold *s (q, $\xi$)*, where $q$ is a high quantile of the series $g$ itself. Here we select $q$ to be the 98[th] percentile of $g$. For these cases, which correspond to days whose SLP map is very similar to that of $\xi$, we can then define exceedances as:

$$u(t,\xi) = g(x(t),\xi) - s(q,\xi)$$

Given $g$ above the threshold, we compute by how much it is so. The cumulative probability distribution *F (u, $\xi$)* of the exceedances thus defined, converges to the exponential member of the Generalised Pareto Distribution (GPD; Freitas *et al.*, 2010; Lucarini *et al.*, 2012). In other words, given a sufficiently long series of SLP maps, we know that the exceedances $u$ computed from these maps obey the following:

$$F(u,\xi) \simeq e^{\left(-\vartheta(\xi)\frac{u(\xi)}{\sigma(\xi)}\right)}$$

Here $\vartheta$ is the extremal index (Moloney *et al.*, 2019), which we calculate using the Süveges Maximum Likelihood estimator (Süveges, 2007), and $u$ and $\sigma$ are parameters of the distribution, which depend on the chosen $\xi$. The local dimension is then obtained as:

$$d(\xi) = \frac{1}{\sigma(\xi)}$$

While the persistence is given by:

$$\theta^{-1}(\xi) = \frac{\Delta t}{\vartheta(\xi)}$$

Where $\Delta t$ is the time interval between successive time steps in our dataset. In practice, we choose each SLP map in the dataset in turn as state of interest $\xi$, which enables us to obtain a value of $d$ and $\theta^{-1}$ for each timestep in our dataset.

In practical terms, $d$ reflects the geometry of the trajectories in a small region (neighbourhood) of the system's phase space around the state of interest. It is therefore related to the number of active degrees of freedom that the system can explore about the state. In other words, it informs on the way the system evolves around the state of interest, and a higher $d$ will correspond
to a less predictable evolution of the system. The persistence $\theta^{-1}$ quantifies for how long the system resides in the neighbourhood of the state of interest. An infinite persistence implies a fixed point of the system, such that all successive timesteps bring no change to the state of the system. At the opposite end of the spectrum, $\theta^{-1} = 1$ corresponds to a non-persistent state of the dynamics. The dynamical systems persistence tends to be very sensitive to small changes in the state of the system. In atmospheric sciences, persistence is often computed as the residence time of the atmosphere within a given cluster of states.
For example, when computing the persistence of weather regimes, one usually counts for how long the atmosphere remains within one given weather regime cluster. However, there are typically a small number of clusters, such that each one contains a relatively large fraction of the total number of timesteps within the dataset. In our case, we define recurrences based on a high threshold such that only a small fraction of timesteps within our dataset qualify as recurrences. By design, $\theta^{-1}$ is thus more sensitive to small changes in the atmosphere than the conventional definition of persistence of weather regimes. The two are
nonetheless related, and relative differences in $\theta^{-1}$ often reflect relative differences in more conventional atmospheric persistence metrics (Hochman *et al.*, 2019).

The above derivations hold under a specific set of conditions, which are seldom satisfied by climate, or indeed any real-world datasets – such as having infinitely long timeseries. However, there are both formal (Caby *et al.*, 2020) and empirical (Messori *et al.*, 2017; Buschow and Friedrichs, 2018) results, which support the application of this framework to natural data. In
particular, Buschow and Friederichs (2018) have shown that $d$ successfully reflects the dynamical characteristics of the atmosphere even for datasets where the universal convergence to the exponential member of the GPD is not achieved. Messori *et al.* (2017) have further shown that the persistence estimates for atmospheric data issuing from the Süveges estimator are very stable under bootstrap resampling of the intra and inter-cluster times (*i.e.* the residence times of the trajectory within and without the neighbourhood of the state of interest). For more details on the estimation of the dynamical systems metrics, the
reader is referred to Lucarini *et al.* (2016) and Faranda *et al.* (2017a; 2019a).

The dynamical systems perspective has been fruitfully applied to a range of geophysical and other datasets (e.g. Faranda *et al.*, 2019b, c; Brunetti *et al.*, 2019; Faranda *et al.*, 2020; Hochman *et al.*, 2020b; De Luca *et al.*, 2020a; Pons *et al.*, 2020). It has also been explicitly shown that $d$ and $\theta^{-1}$ can offer an objective characterization of synoptic systems over different geographical regions, including the Mediterranean (Hochman *et al.*, 2019; De Luca *et al.*, 2020b), the North Atlantic (Faranda *et al.*, 2017a; Messori *et al.*, 2017; Rodrigues *et al.*, 2018) and the full Northern Hemisphere (Faranda et al., 2017b). In this study, we compute $d$ and $\theta^{-1}$ for daily and 6-hourly 500 hPa geopotential height (Z500) and SLP fields from the NCEP/NCAR reanalysis, over the Eastern Mediterranean, placing Israel in the middle of the domain (27.5°N-37.5°N; 30°E-40°E; Fig. 1). To understand the differences between heat waves and cool days, we analyse both the CDFs (Cumulative Distribution Functions) and the mean temporal evolution of the two groups of days in terms of $d$ and $\theta^{-1}$. The Wilcoxson Rank-Sum (comparing the medians) and Kolmogorov-Smirnov (comparing the CDFs) tests are used for estimating the differences between the upper and lower 10% of CSI days at the 5% significance level. A bootstrap sampling test with $10^4$ samples is used to evaluate the 95% confidence intervals of the mean temporal evolutions.

Previous studies have shown that the dynamical systems metrics $d$ and $\theta^{-1}$, have a strong seasonal cycle (Faranda *et al.*, 2017a, b; Rodrigues *et al.*, 2018; Hochman *et al.*, 2020a). Thus, we remove the seasonal cycle before comparing the various events. Since we are comparing individual days/events during different parts of the summer season, it is better to de-seasonalize the data in order to study the anomalies. In other words, we test whether heat waves are synoptically and dynamically unusual with respect to the cool days in the same part of the season. The seasonal cycle is estimated by averaging the metrics for a given time step (e.g., 15 August at 12UTC) over all years, repeating this for all time steps within the year and ultimately smoothing the series with a 30-day moving average.

## 2.4 Forecast spread/skill

To obtain an ensemble forecast, a set of numerical forecasts are performed with either different initial conditions, and/or perturbation of physical parametrizations. Ensemble forecasts offer an efficient way of estimating uncertainty by computing the ensemble spread. This is quantified by estimating the standard deviation between ensemble members. The spread can be taken as an indicator of practical predictability: in a perfect ensemble, a small spread would generally indicate we can determine with a good degree of confidence the future weather, while a large spread would point towards a larger uncertainty (e.g., Buizza 1997). This type of approach is commonly used when investigating atmospheric predictability (e.g., Hohenegger *et al.*, 2006; Ferranti *et al.*, 2015), although it does have limitations (e.g., Whitaker and Lough, 1998; Hopson 2014).

An additional frequently used forecast diagnostic is the absolute error, which provides a measure of forecast skill. The correlation between the ensemble spread and skill of the NWP model indicates the extent to which the ensemble can be used to provide an *a priori* estimate of the practical predictability of the atmospheric configuration we are considering. Here, we use the homogeneous station archive mentioned in Sect. 2.1 above as ground truth to estimate the forecasts' absolute error. In

order to remove biases due to topographic differences between the model and the stations, the GEFS reforecast gridded data is bilinearly horizontally interpolated to the location of the stations. The bias computed over the whole period is then removed for each station.

The GEFS reforecasts are initialized at 00UTC and are available at three-hour intervals. Since our analysis focuses on heat waves, we estimate the spread/skill for maximum temperature and SLP at a lead-time of 69 hours, while the maximum temperature is defined between 45 h and 69 h. Given the three-hour interval of the forecast data, and bearing in mind that each station's maximum temperature is recorded between 20UTC and 20UTC of the next day, this time-window roughly corresponds to the definition of maximum temperature for the station data. Since the dynamical systems metrics offer information on the temporal evolution of the atmosphere in the neighbourhood of a given reference state, we argue that using the time of forecast initialization as temporal coordinate when plotting spread and error is most indicative for comparing the dynamical systems and numerical forecasts. In the supplementary material, we also plot the spread/skill for the forecasts initialised 69 h before the marked time. Thus, the plots in the main text show forecast initialisation times, while those in the supplementary material show the forecast valid times. Statistical inference is accomplished by the same tests mentioned in Sect. 2.3.

## 2.5 Air parcel tracking

In order to identify typical pathways of air masses leading to situations with high and low CSI values, ten-day backward trajectories are computed using the Lagrangian Analysis Tool (LAGRANTO; Wernli and Davies, 1997; Sprenger and Wernli, 2015). The reader is referred to Fig. 2 in Sprenger and Wernli (2015) for a schematic overview of the typical steps taken to compute trajectories. The tracking of temperature and potential temperature along the trajectory further allows to quantify the contribution of adiabatic and diabatic processes to the anomalous temperatures. The vertical and horizontal wind components required for the trajectory computations are acquired from the ERA-Interim reanalysis (Dee *et al*. 2011, Sect. 2.1). The trajectories are initialized at 12UTC from fixed points in the whole study region on the first day of a heat wave or cool days (Fig. 1). In order to analyze the near-surface air masses, i.e. those related to the hot and cool conditions, we consider trajectories that are initialized between the surface and 90 hPa above the surface. According to recent studies, the planetary boundary layer height in Israel during summer is ~600 – 900 m above the surface (Uzan *et al.*, 2016; 2020). Assuming hydrostatic balance and thus a pressure decrease of approximately 1 hPa every 8m height difference, 90 hPa corresponds to about 720m. Therefore, this can be considered a reasonable choice.

The trajectories are calculated from 6-hourly ERA-Interim data and remapped to a 1° regular latitude-longitude grid. Thus, the analyzed wind field does not resolve sub grid-scale processes, such as Lagrangian transports due to small convective cells. Also, vertical motion associated with short-lived convection between two-time steps is not accounted for. Still, for a climatological investigation that is the focus of this study, the trajectory calculation is a suitable diagnostic.

## 3.  Results

### 3.1 Dynamics of heat waves over the Eastern Mediterranean

We first analyze the differences between heat waves (upper 10% of CSI values) and cool days (lower 10% of CSI values). From an atmospheric dynamics' standpoint, the main difference between the two groups is that heat wave days are associated with an upper level ridge, whose center is located in the south – eastern part of the study region (Fig. 1a), while cool days are associated with an upper level trough, whose center is located at the north – western part of the study region (Fig. 1b). The SLP patterns are quite similar in both groups, but the heat waves show lower SLP in the south-west and a higher SLP in the north-east compared to the cool days sample (Fig. 1c). This implies stronger pressure gradients in the cool days' subgroup, leading to enhanced cool air advection from the Mediterranean Sea inland, in comparison to the heat wave days. Furthermore, the above reveals that the large-scale configuration is an important factor in the generation of a heat wave over the Eastern Mediterranean. The backward trajectory air parcel analysis illustrates that the flow preceding an extreme heat wave has a roughly meridional orientation when traveling over the Eastern Mediterranean and originates over the European continent (Fig. 2a). On the other hand, the air parcels for cool days often originate over the Atlantic, and take a more zonal pathway across the Eastern Mediterranean (Fig. 2b). The initial potential temperature of the heat wave air masses is about 7 K higher than that for the cool days (Fig. 2e). The differences in potential temperature between the two groups can mainly be attributed to the more continental origin of the air parcels for the heat waves, thus potentially transporting warmer air masses that descend on their path to the target region. Their descent, which is stronger than for cool days (Fig. 2c), is accompanied by a temperature increase of more than 25 K during the ten-day period (Fig. 2d). The potential temperature remains nearly constant until the final stages of the descent except for the diurnal cycle (Fig. 2e). Thus, we conclude that the extreme heat is related to an adiabatic descent of the air parcels over the Eastern Mediterranean rather than to diabatic heating. In other words, the warm air parcels are transported towards the Eastern Mediterranean with the governing westerlies rather than heated up locally over several days. This supports the findings of Harpaz *et al*. (2014), who argued that extreme summer heat waves over the Eastern Mediterranean are mostly regulated by mid-latitude disturbances rather than by the Asian Monsoon, as previously proposed by Ziv *et al*. (2004). An additional important difference between the two sets of CSI events is that, unlike for the heat waves (Fig. 2a, f), the specific humidity of the cool days increases by 2 g kg$^{-1}$ around t = -48 hours, due to the longer stretch the latter air parcels follow over the Mediterranean Sea (Fig. 2b, f) and perhaps some enhanced convection, which our analysis does not account for. Indeed, comparing the portion of terrestrial back-trajectories for heat waves and cool days (Fig. S2) suggests that for most of the time they are quite similar. It is only 72-24 hours prior to the events that the portion of terrestrial back-trajectories for cool days reaches a minimum and is much lower than for heat waves (Fig. S2).

From a dynamical systems point of view, the upper and lower 10% of CSI also exhibit substantial differences. Fig. 3 shows a phase-plane diagram for *d* and *θ* computed on Z500 and SLP for the heat waves and cool days. *θ* is significantly lower at both levels for heat waves with respect to cool days, i.e., the former are generally more persistent systems. Statistically significant differences in the median local dimensions (*d*) of the two groups are found only for the Z500 variable, for which the heat

waves typically display a lower local dimension ($d$) than the cool days (Fig. 3a). The clear separation between the two groups, especially at upper level (cf. Fig. 3a and Fig. 3b) correlates well with the atmospheric dynamics' viewpoint, which also shows more pronounced differences at Z500 (Fig. 1). This points to the importance of using different variables at different pressure levels to obtain a comprehensive picture of the dynamics of heat waves.

Fig. 4 displays the average temporal evolution of $d$ and $\theta$ during the selected events, again computed for Z500 and SLP. Zero on the x-axis denotes the first day of the event at 12UTC, whereas, a value of zero on the y-axis implies that the events are not different from the climatology of the days they occurred in. Substantial differences are found between the time evolutions of the upper and lower 10% of the CSI events. For Z500, the temporal evolution of $d$ and $\theta$ for heat waves are in phase with each other, and show a minimum with below climatology values in the 24 h preceding the event onset (Fig. 4a). While there is still a considerable spread around the mean, even the upper bounds of our confidence intervals are well below zero in the build-up to the events. Instead, cool days display weak positive anomalies of $d$ and $\theta$, but these are almost never significantly different from 0 (Fig. 4b). The dynamical systems metrics computed on SLP provide a completely different picture: heat waves typically display a weak above-climatology $d$, which increases towards the event onset and then decreases (Fig. 4c). $\theta$ displays a slightly below-climatology persistence (i.e. positive anomalies) and decreases towards the event onset (Fig. 4c). However, the very large spread in the composite evolution, and in particular in $d$, suggests some caution in over-interpreting the details of these evolutions. Cool days are characterized by higher positive anomalies of $d$ and $\theta$ in the days preceding the event. The build-up towards this type of event is characterized by an increase in $\theta$ (decrease in persistence) and a decrease in $d$ (decrease in active degrees of freedom; Fig. 4d). The cool days also appear to have a more coherent evolution (lower spread around the mean) than the heat waves for SLP.

The differentiation between the two samples is thus more pronounced when computing the metrics on Z500 than on SLP (Fig. 4), as also shown in the daily distributions (Fig. 3). Moreover, the variability in the temporal evolution of the dynamical systems metrics is smaller in Z500 than in SLP (Fig. 4). This points to: i) coherent, and very different, upper level conditions, which engender the two sets of CSI days; and ii) a comparatively wide range of possible near-surface patterns leading to severe heat waves. The latter may be explained by the fact that, given initially warm upper-level air parcels and upper-level subsidence leading to rapid adiabatic warming, the occurrence of a heat wave is then relatively insensitive to the details of the surface conditions (e.g., Baldi *et al.*, 2006; Harpaz *et al.*, 2014). Our general understanding of the synoptic conditions at surface levels further suggests that the delicate interplay between the Persian Trough and Subtropical High systems (Alpert *et al.*, 1990) may contribute to the large spread of both heat waves and cool days regarding the dynamical systems metrics computed on SLP.

We analyze next numerical ensemble forecasts from the GEFS reforecast dataset for both sets of events. Substantial differences are again found between the two groups (Fig. 5). Both the ensemble spread and the absolute error are significantly higher for heat waves than for cool days (Fig. 5). The model spread and absolute error increase before the onset of the heat wave, peaking at around 24-48 hours negative lags (Fig. 5). This pattern stands in stark contrast to the temporal evolution of $d$ computed on Z500 (cf. Fig. 5a, c, e and Fig. 4a), but somewhat resembles the evolution of $d$ computed on SLP, albeit with a ~24 hours shift

in time (cf. Fig. 5e and Fig. 4c). Such a shift may be explained by the fact that the spread/skill of the ensemble forecasts are computed every 24 hours, while the dynamical systems metrics are instantaneous in time (local in phase-space) and computed from 6-hourly data. The reforecasts computed for the individual stations (not shown) resemble the average forecast spread/skill (Fig. 5). The corresponding plots for forecast valid time (see Sect. 2.4), are provided in Fig. S3.

### 3.2 Analysis of the mid-August 2010 heat wave over the Eastern Mediterranean

The mid-August 2010 heat wave over the Eastern Mediterranean was a severe heat wave, which lies in the upper 0.3% of the CSI distribution. A detailed analysis of the heat wave highlights both similarities and differences with the climatology of the heat wave days (Sect. 3.1). The Z500 and SLP patterns for 15[th] August 2010 are comparable with the average configuration of
a heat wave, but show a stronger upper level ridge and meridionally-oriented isobars (cf. Fig. 6a and Fig. 1a). From a dynamical systems point of view, the 2010 heat wave was also an uncommon extreme, especially for the metrics computed on Z500. The dynamical systems metrics' anomalies computed on this field range between -0.9 and -1.4 for $d$, and -0.14 and -0.2 for $\theta$ (Fig. 6b). This situates the heat wave in the lower 10% of the respective distributions (see also red dots in Fig. 3a). During its evolution, the event displays an increase in both $d$ and $\theta$ computed on Z500 and a decrease (increase) in $\theta$ $(d)$ computed on
SLP (Fig. 6b, c). While the Z500 $d$ and $\theta$ evolution is roughly comparable to that identified for heat wave days (cf. Fig. 4a and 6b), the SLP $d$ and $\theta$ evolutions show profound differences. This may simply reflect the larger spread in dynamical systems properties across the different heat waves for SLP than for Z500, which is likely to be partially modulated by interactions between the surface and the atmosphere. Naturally, these interactions predominantly affect the lowermost parts of the Troposphere. We further hypothesize that differences between the single case and the climatology may be related to the
relatively small day to day variations during summer over the Eastern Mediterranean, which make it challenging to depict the exact onset of a heat wave. Indeed, when comparing the climatology of the temporal evolution of $d$ and $\theta$ for Z500 (Fig. 4a) with the single case (Fig. 6b) it is relatively easy to see that there is an increase in both $d$ and $\theta$ as the heat wave develops. On the other hand, when comparing the temporal evolution of $d$ and $\theta$ for SLP (Fig. 4c with Fig. 6c), one can see that depicting the exact time the heat wave starts is very important for comparison. In both Figures, $d$ increases and $\theta$ decreases at some point
in the chosen time window, but the timing of these trends is shifted between the climatology (Fig. 4c) and the single case (Fig. 6c).

The 2010 heat wave was also uncommon in terms of the large-scale flow and Lagrangian trajectories (Fig. 7). Between -10 to -5 days prior to the event, the majority of air parcels were transported in an easterly flow on the southern flank of an anticyclone located over Russia. Thus, air parcels came from the Zagros Plateau of Northern Iran, rather than from central Europe as in
the climatology (cf. Fig. 7a, b and Fig. 2a). Indeed, Zaitchik *et al*. (2007) have argued that the Zagros Plateau has a strong influence on extreme summertime heat waves over the Eastern Mediterranean. Here we show that the anti-cyclonic wave breaking of the blocking regime over Russia, which interestingly is related to the decay phase of the Russian 2010 heat wave

(Quandt *et al.*, 2019), played an important role in transporting the warm air masses from Northern Iran towards the Eastern Mediterranean and Israel (Fig. 7a, b and Fig. 8). This is realized through the trough east of the blocking ridge centered over European Russia, which is tilted southwest - northeast and advected westward (Fig. 8b - d, Davini *et al.*, 2012; Quandt *et al.*, 2019). For the last five days prior to the heatwave (Fig. 7b), the parcel's trajectories resemble more closely the climatology of heat waves (Fig. 2a). Reflecting the different advection pathways, the initial potential temperature and temperature of the air parcels are respectively about 2K and 7K higher than the climatology of heat waves (cf. Fig. 7d, e and Fig. 2d, e). Accordingly, the hot air masses in the mid-August 2010 heat wave are transported to the Eastern Mediterranean and undergo adiabatic heating, rather than being heated up locally. This is in line with the climatology discussed in Sect. 3.1, and heat waves in other parts of the world (e.g., Bieli *et al.*, 2015; Quinting and Reeder, 2017; Zschenderlein *et al.*, 2019).

Fig. 9 shows the temporal evolution of the forecast spread/skill for the mid-august 2010 heat wave compared to the heat wave climatology. Throughout the lead up and early phases of the event, the forecast displays a lower spread and error than for other heat waves. A large decrease in the practical predictability occurs as the event develops, i.e., an increase in the spread/skill for maximum temperature (Fig. 9a, b). This mirrors the increase in $d$ and $\theta$ computed on Z500 and for $d$ computed on SLP (cf. Fig. 9a, b with Fig. 6b, c). Indeed, the decay phase of the Russian heat wave was characterized by low practical predictability (Matsueda 2011), which may have influenced the predictability over the Eastern Mediterranean. However, it should be noted that the spread computed on maximum temperature for the mid-August 2010 heat wave does not correlate well with the spread computed on SLP (cf. Fig. 9a with Fig. 9c). Moreover, some striking differences are displayed between the ensemble forecast of this single event and the climatology of forecasts for heat waves. These discrepancies may be related to the fact that we are analyzing a single event, whose error may not reflect the practical predictability of the atmosphere even for a perfect ensemble (e.g., Whitaker and Lough, 1998; Buizza *et al.*, 2005; Hopson 2014). The corresponding plots for forecast valid time (see Sect. 2.4), are provided in Fig. S4.

## 4. Summary and conclusions

Heat waves are a major weather-related hazard, especially in an era of rapid climate change. We define heat waves over the Eastern Mediterranean according to a state-of-the-art 'Climatic Stress Index' (CSI; Saaroni *et al.*, 2017), developed specifically for the region's summer weather conditions. We use a combination of dynamical systems theory, numerical weather forecasts and air parcel back-trajectories to investigate the evolution and predictability characteristics of heat waves (high CSI) and cool days (low CSI) for the region.

The main conclusions are as follows: significant differences are found between heat waves and cool days from both a dynamical systems and numerical weather prediction perspectives. Heat waves show relatively low practical predictability (large model spread/low skill) in the ensemble reforecast dataset used here, in spite of the relatively stable flow characteristics (high intrinsic predictability) compared to the cool days. When considering Z500, the *intrinsic* predictability of heat waves

over the Eastern Mediterranean is highest, i.e., low local dimension ($d$) and high persistence (low $\theta$), in the 24 h preceding the onset of the event, and lowest in the decay phase of the event. Indeed, Lucarini and Gritsun (2020) recently argued that atmospheric blocking over the Atlantic also displays such characteristics. The persistent upper level ridge that characterizes the heat waves may explain the high intrinsic predictability during the onset phase. The dynamical systems metrics computed on SLP show a different temporal evolution to their Z500 counterparts, emphasizing the different characteristics of the atmospheric flow at the different vertical levels. Specifically, there is a very large spread across different heat wave events. We argue that this may be associated with the delicate interplay between the Subtropical High and the Persian Trough at surface levels (Alpert et al., 1990), which can lead to a range of different SLP configurations all leading to a heat wave. This may further be reasonably attributed to the interactions between the land/sea surface and the atmosphere, which mainly influence the lower parts of the Troposphere. However, it is important to note that in many – albeit certainly not all – cases these interactions influence the atmosphere at time scales longer than those we consider in our analysis (e.g., Entin et al., 2000), and act as a seasonal-scale preconditioning to extremely high summer temperatures (Zampieri et al., 2009; Zittis et al., 2014).

Based on the Lagrangian air parcel analysis, we conclude that the physical processes governing Eastern Mediterranean summer heat waves relate to adiabatic descent of the air parcels over the region rather than diabatic heating, in agreement with previous findings (e.g., Harpaz et al., 2014). In other words, the air parcels are transported horizontally and vertically towards the Eastern Mediterranean with the governing westerlies rather than heated up locally over consecutive days. We further conclude that the origin of the air parcels over land in the days before the onset of a heat wave plays an important part in its generation.

A detailed analysis of the record-breaking mid-August 2010 heat wave provides further insights in this respect, by underscoring how the parcels, which contributed to the heat wave, were warmer than those of the climatology of heat waves as early as 10 days prior to the event. Interestingly, the onset of the heat wave over the Eastern Mediterranean was related to the decay phase of the Russian heat wave (Quandt et al., 2019) and we conclude that the anti-cyclonic Rossby wave breaking over Russia contributed to the onset of the Eastern Mediterranean heat wave. The 2010 heat wave showed both differences and similarities to other heat waves, highlighting the range of possible atmospheric and dynamical developments leading to high CSI values. This is compounded by the general difficulty of analyzing the life-cycle of heat waves, since there is little agreement as to what exactly a heat wave is and when it starts and ends (Shaby et al., 2016).

We conclude that the instantaneous dynamical systems metrics of local dimension ($d$) and persistence ($\theta^{-1}$) provide complementary information on extreme summer heat waves compared to the conventional analysis of numerical weather forecasts. The discrepancy between the practical and the intrinsic predictability of the heat waves reflects this complementarity. For example, we interpret a very persistent system as being intrinsically highly predictable, yet the numerical forecasts we analyse display larger spread and error for the more persistent atmospheric configurations. In this respect, having an a priori measure of the persistence of an atmospheric configuration from dynamical systems can be a useful complement to the numerical forecast. Specifically, the practical predictability relies on the performance of a numerical forecast model. As such

it blends model and data assimilation biases with the intrinsic characteristics of the atmospheric flow. Moreover, even a perfect ensemble may not provide a good skill-spread relationship (Hopson 2014). That is, even a perfect ensemble may have a spread that does not always reflect the actual forecast error (Whitaker and Lough, 1998). In the specific case of the heat waves we analyze here, the spread and skill were well correlated for maximum temperature, but this is not a universal rule. For example, the mid-august 2003 heat wave had a very low spread (Tmax spread = 0.2 K, cf. Fig. 5b) and an above average error (Tmax absolute error = 1.1 K, cf. Fig. 5d). However, both $d$ and $\theta$ computed on Z500 display a strong increase ($d$ = -1.7 to 0.3, $\theta$ = -0.2 to 0.1 over the considered time window, not shown; cf., Fig. 4a and Fig. 6b), pointing to a decrease in intrinsic predictability. In such cases, local dimension ($d$) and/or persistence ($\theta^{-1}$) trends that seem to contradict a low ensemble spread may serve as a warning of a potentially poor spread-skill relationship.

As a caveat, the comparison of the practical and intrinsic predictability still carries some interpretation challenges. Although the differences between the two can be partly ascribed to the different characteristics of the two measures, they may also be subject to the shortcomings of the GEFS ensemble data. In particular, the spread of the GEFS ensemble data, as most NWP ensemble forecasts, does not always reflect the practical predictability of the atmospheric flow (e.g., Whitaker and Lough, 1998; Keune *et al.*, 2014). Moreover, our interpretation of the dynamical systems metrics may also be imperfect. Indeed, the metrics provide local information in phase space, while the spread and error of an ensemble forecast presumably reflect the longer-term evolution of the atmospheric flow. Similar interpretation challenges for the practical vs. intrinsic predictability have emerged when studying cold spells over the Eastern Mediterranean (Hochman *et al.*, 2020a).

Notwithstanding these ongoing challenges, we believe that the novel view presented here, which leverages a dynamical systems approach for diagnosing extreme weather events, outlines an important avenue of research. We trust that it may be successfully applied to other regions and weather extremes in the future.

**Acknowledgements**

We would like to thank Mr. Yitzhak Yosef, head of the climate research department at the Israeli Meteorological Service (IMS), for providing homogeneous high-quality continuous station data over Israel. We further thank Dr. Noam Halfon of the IMS for providing averaged summer temperatures computed from the IMS Climatic Atlas. We thank the National Centers for Environmental Prediction/National Center for Atmospheric Research for the Reanalysis data and the Global Ensemble Forecast System reforecasts. We are grateful to Heini Wernli and the Atmospheric Dynamics group at ETH Zurich for sharing the ERA-Interim data. This paper is a contribution to the Hydrological cycle in the Mediterranean Experiment (HyMeX) community.

**Funding**

AH is funded by the German Helmholtz Association (ATMO Program). The contribution of JQ was funded by the Helmholtz-Association (grant VH-NG-1243). JGP thanks AXA Research Fund for support (https://axa-research.org/en/project/joaquim-pinto). GM was partly supported by the Swedish Research Council Vetenskapsrådet (grant no. 2016-03724).

**Author contributions**

All authors have contributed to the conceptual development of the study. AH and GM analyzed the data from a dynamical systems perspective. SS analyzed the forecast model data. JQ computed the air parcel backward trajectories. AH drafted the first version of the manuscript. All authors contributed through discussions and revisions.

**Data availability**

The paper and/or the supplementary materials contain or provide instructions to access all the data needed to evaluate the conclusions drawn in the paper. Additional data is available from the corresponding author upon request.

**Competing interests**

The authors declare no competing interests.

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

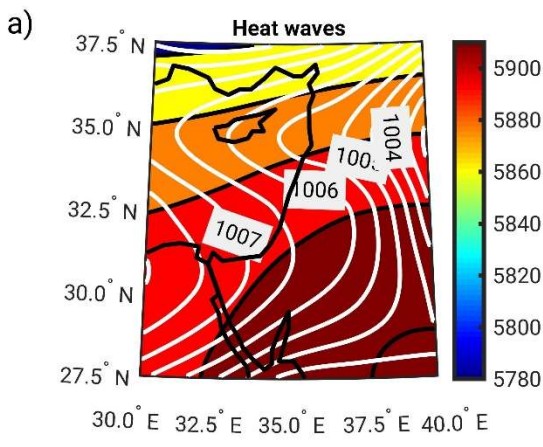

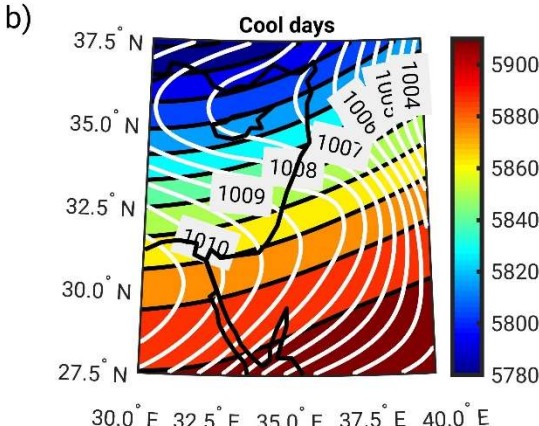

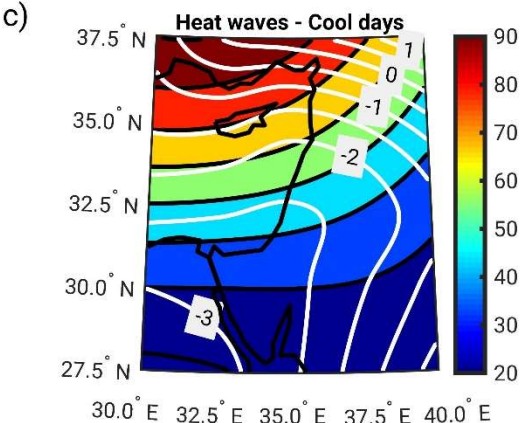

**Figure 1** Mean sea level pressure (SLP in hPa, white contours) and 500 hPa geopotential height (Z500 in m, shaded in color) for the 10% of days with the highest (heat waves) and lowest (cool days) Climatic Stress Index (CSI) values. (a) Heat wave days mean composite; (b) cool days mean composite; (c) heat waves minus cool days.

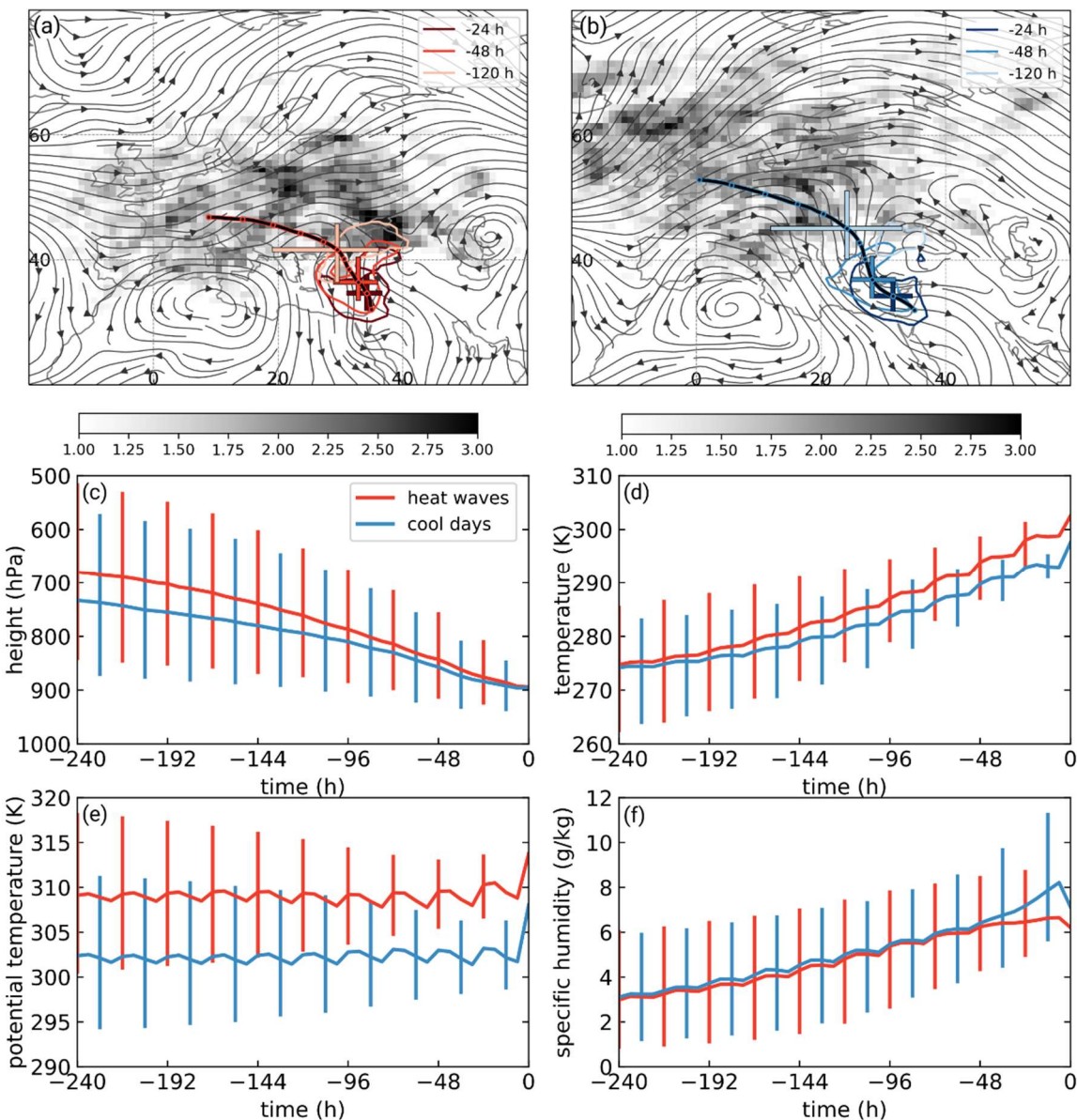

**Figure 2** Median backward trajectory for (a) heat waves (upper 10% of CSI) and (b) cool days (lower 10% of CSI), with circles indicating days (from 10 days before onset to onset). Grey shading show trajectory density 10 days before onset (number of trajectories per 1000 km²), while contours show trajectory density for the indicated time lags (5, 2, 1 days before onset, contours denote a density of 20 trajectories per 1000 km²). Streamlines of 800 hPa winds averaged between -5 to -1 days are included. The inter-quartile range of trajectory positions is shown with crosses for the different time lags. Median evolution of

(c) pressure (hPa); (d) temperature (K); (e) potential temperature (K); and (f) specific humidity (g kg⁻¹) of air parcels. Heat waves are indicated in red and cool days in blue. The inter-quartile range is plotted for the physical properties of the air parcels. 0 h corresponds to the first day of CSI ≥ 90% or CSI ≤ 10% and at 12UTC.

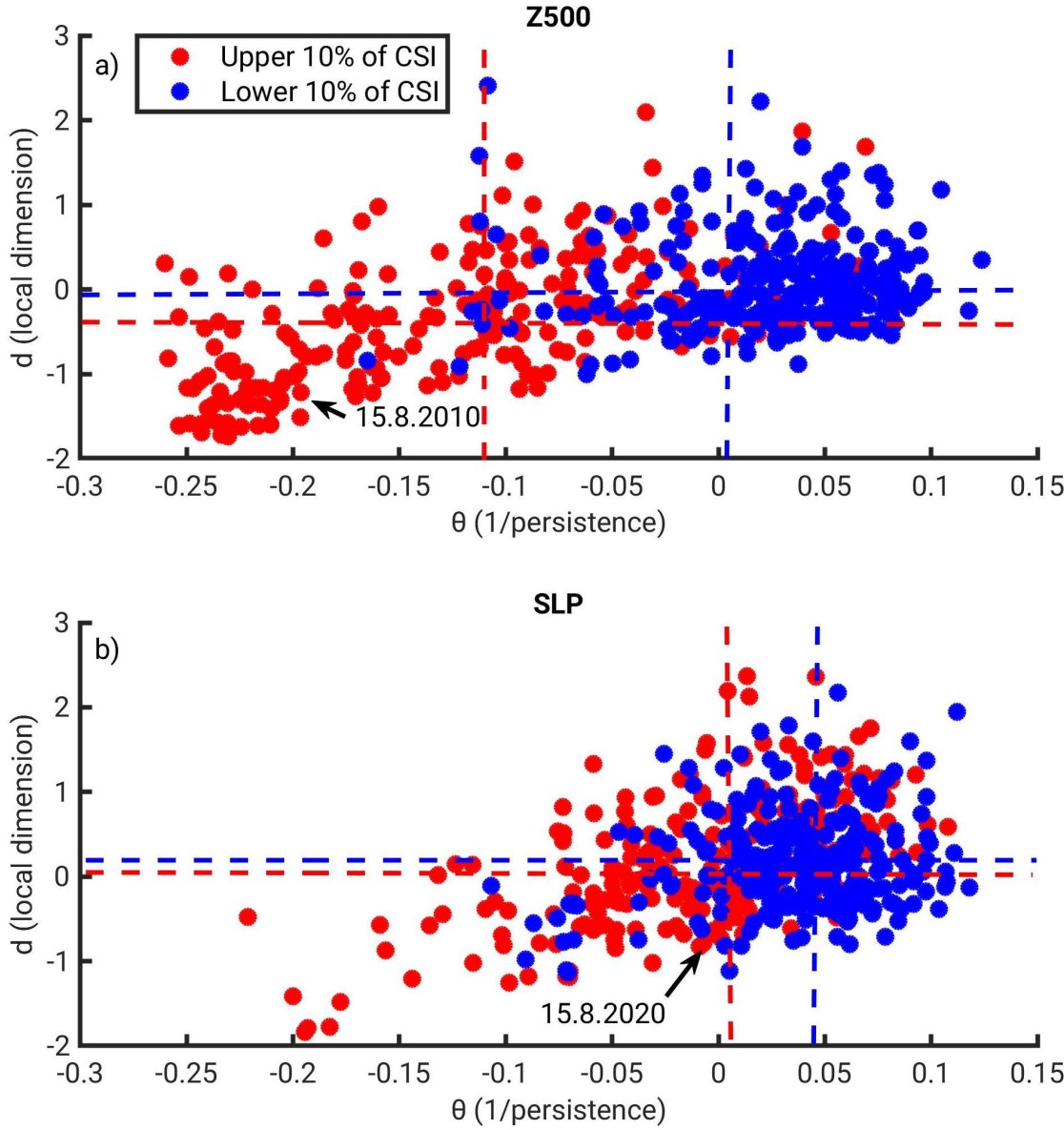

**Figure 3** A phase-plane diagram for the upper and lower 10% of CSI daily values (heat waves in red and cool days in blue). The de-seasonalized dynamical systems metrics (*d* and *θ*) were computed for: (a) Z500 and (b) SLP. Dashed lines represent the median values of *d* and *θ*. The 15.8.2010 is indicated by the black arrows.

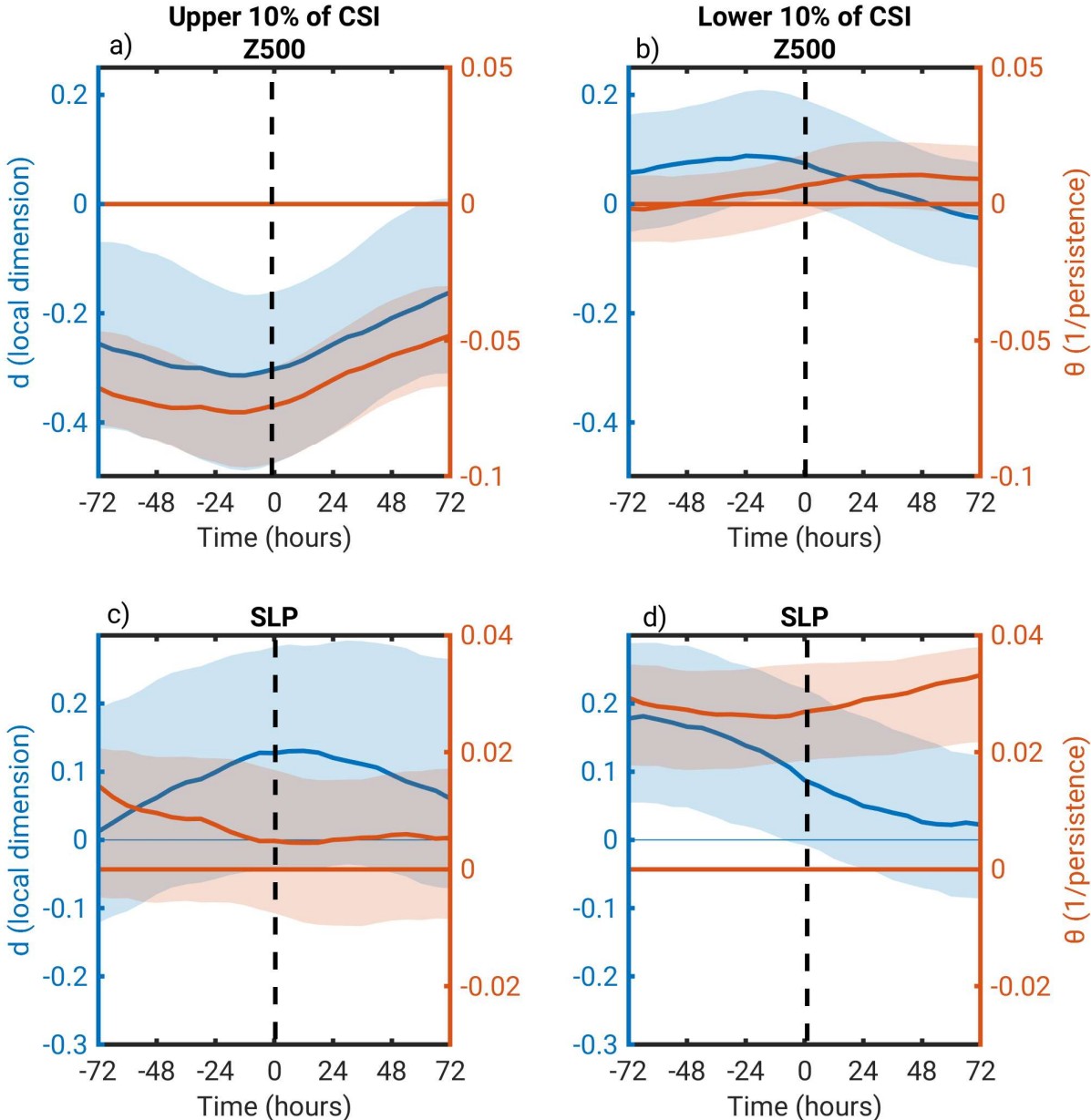

**Figure 4** The average temporal evolution of the dynamical systems metrics (*d* and *θ*) for heat waves (upper 10% of CSI) and cool days (lower 10% of CSI) events. The dynamical systems metrics were computed for: (a, b) Z500 and (c, d) SLP. The events are centered (0 h) on the first day of CSI ≥ 90% or CSI ≤ 10% and at 12UTC. A 95% bootstrap confidence interval is plotted in shading.

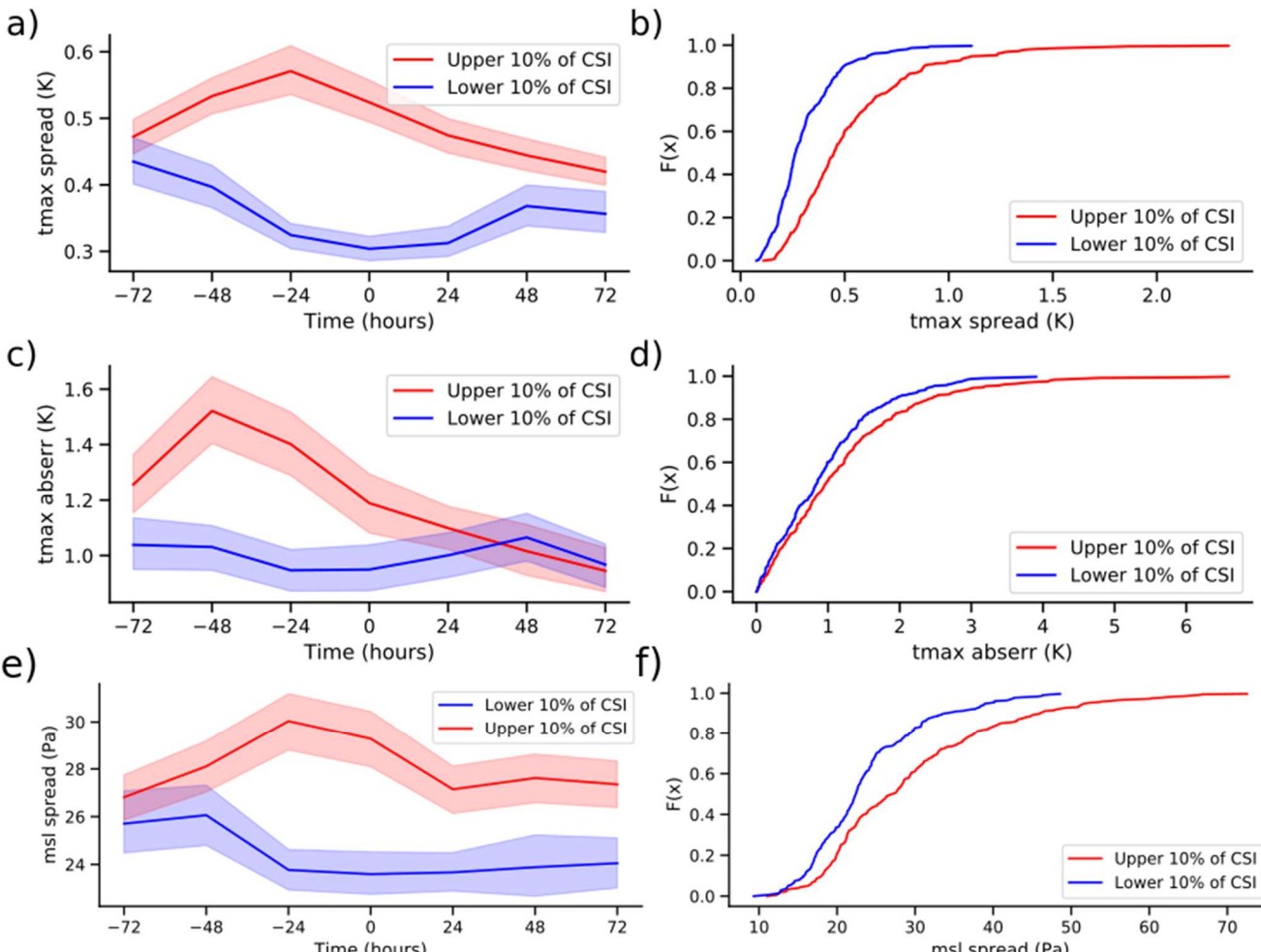

**Figure 5** Forecast spread/skill for heat waves (upper 10% of CSI) and cool days (lower 10% of CSI). The lines show the mean temporal evolution of the ensemble model spread for Tmax (a), SLP (e) and absolute error for Tmax (c) of forecasts with lead-time 69h, initialized at different time lags with respect to the events, calculated every 24 hours. The events are centered (0 h) on the first day of CSI ≥ 90% or CSI ≤ lower 10% and at 12UTC. The CDFs of the mean ensemble forecast model spread for Tmax (b), SLP (f) and absolute error of Tmax (d) for the forecasts with lead-time 69h initialised at 00UTC. A 95% bootstrap confidence interval is shown in shading for the temporal evolution plots (a, c, e). The plots are displayed for the time of forecast initialization (see Sect. 2.4).

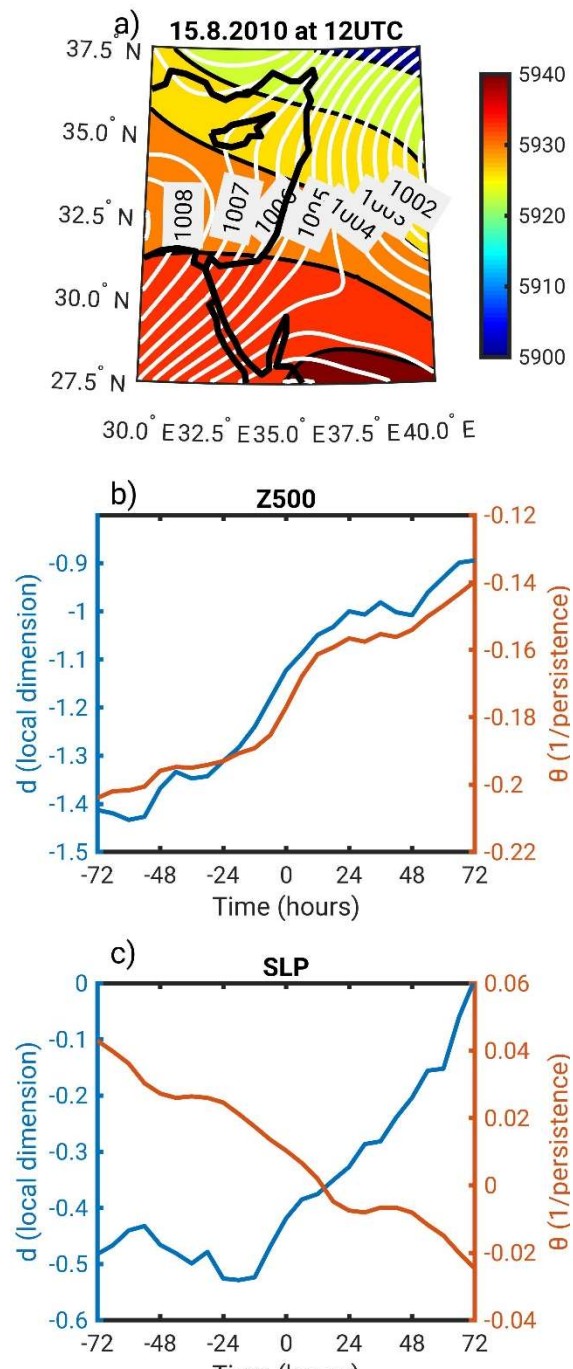

**Figure 6** A dynamical systems analysis for the mid-august 2010 heat wave. (a) SLP (white contours in hPa) and Z500 (shading in m) on 15.8.2010 at 12UTC. The dynamical systems metrics' (*d* and *θ*) temporal evolution centered on 15.8.2010 at 12UTC (0 h) computed on (b) Z500 and (c) SLP.

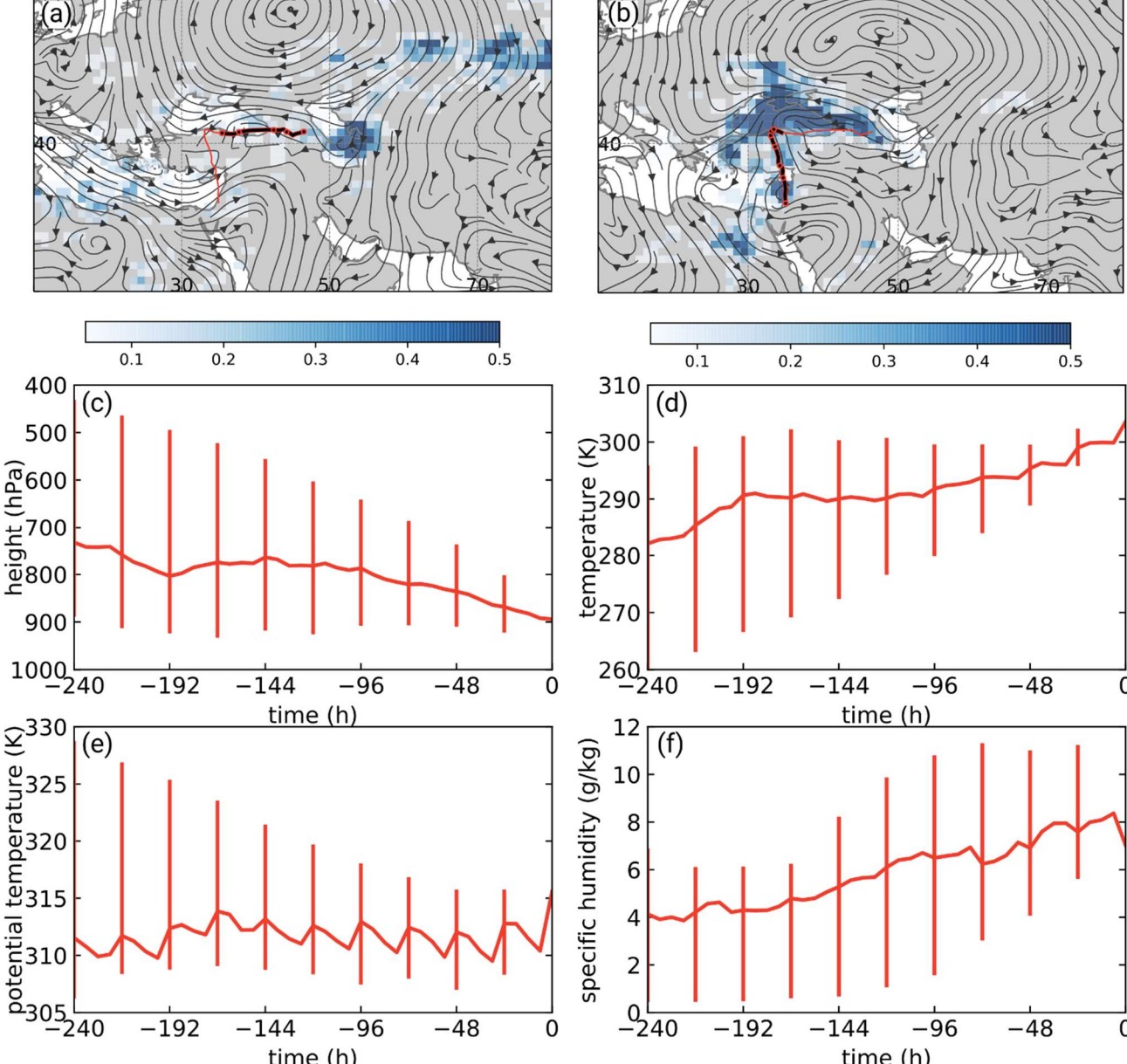

**Figure 7** Backward trajectory air parcel tracking for the mid-August 2010 heat wave initialized on 15.8.2010 at 12UTC with (a) circles indicating days (from -10d to -6d before 15.8.2010 at 12UTC), grey shading indicating trajectory density 10 days before onset (number of trajectories per 1000 km²), and stream lines of 800-hPa wind (averaged between -10d to -6d before 15.8.2010 at 12UTC). (b) as in (a), but for -5d to -1d and trajectory density 5 days before onset. Median evolution of (c) height (hPa); (d) temperature (K); (e) potential temperature (K); and (f) specific humidity (g kg[-1]) of the tracked air parcels. The inter-quartile range is plotted for the physical properties of the air parcels. 0 h corresponds to 15.8.2010 at 12UTC.

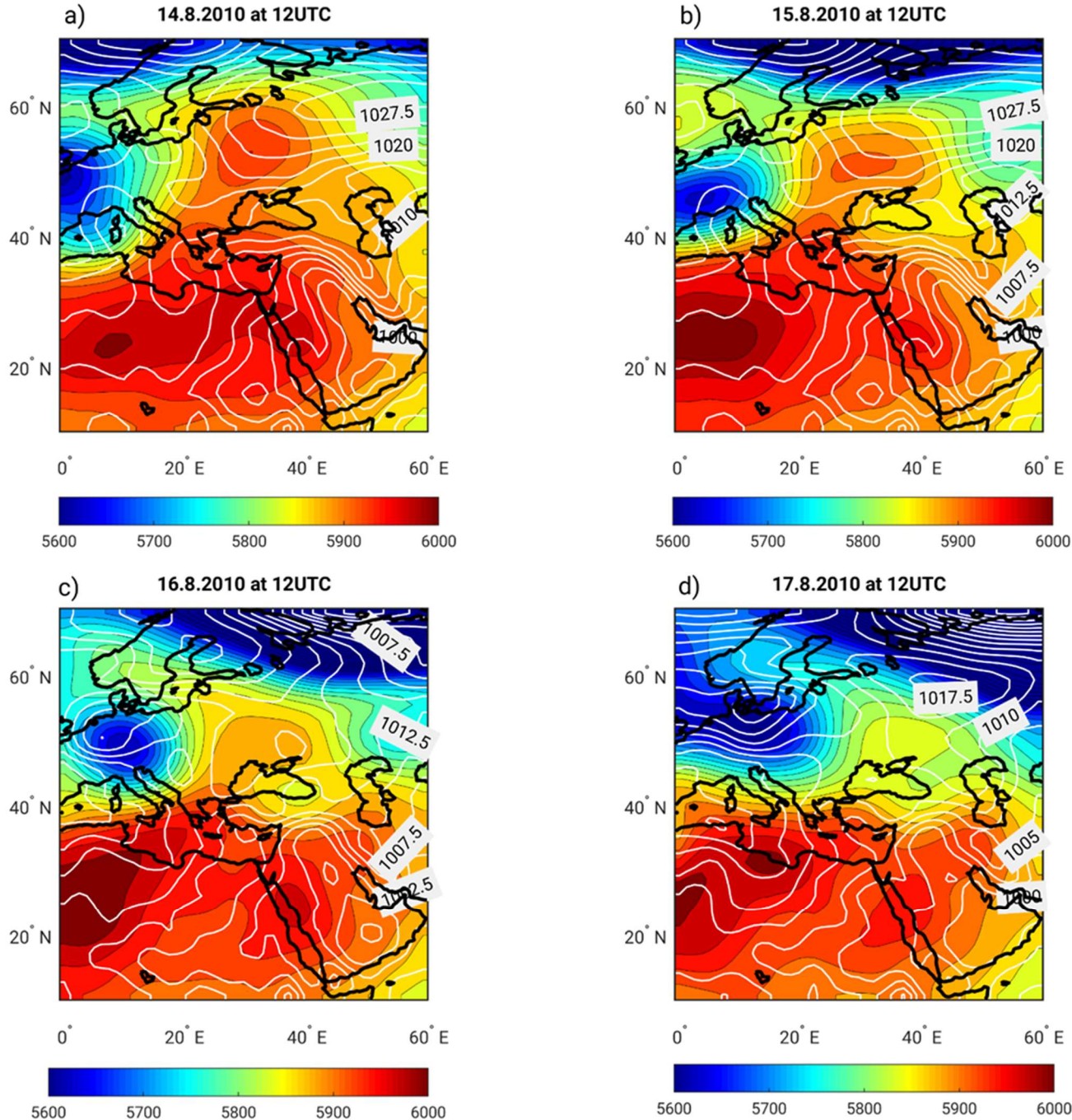

**Figure 8** The large-scale evolution of SLP (white contours in hPa) and Z500 (shaded color in m) for the mid-August 2010 heat wave. a) 14.8.2010 at 12UTC; b) 15.8.2010 at 12UTC; c) 16.8.2010 at 12UTC; and d) 17.8.2010 at 12UTC.

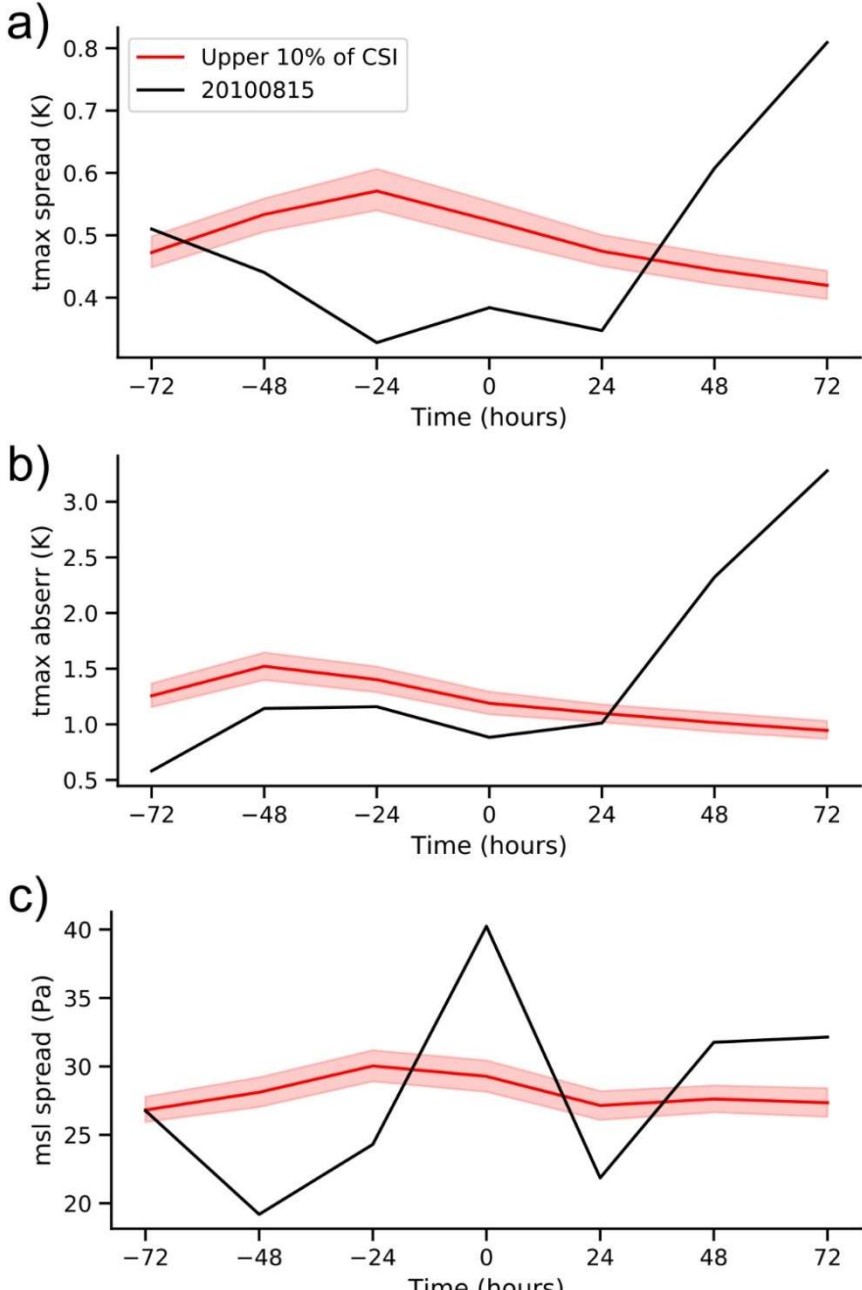

**Figure 9** Forecast spread/skill for the mid-August 2010 heat wave, centered (0 h) on 15.8.2010 at 12UTC (black line). The mean temporal evolution of the ensemble model spread for Tmax (a), SLP (c) and absolute error for Tmax (b) of forecasts with lead-time 69h, initialized at different time lags with respect to the event, computed every 24 hours. The heat waves (upper 10% of CSI - red lines) are displayed for reference. A 95% bootstrap confidence interval for all heatwaves is displayed in shading.