# Peer review of "A New View of Heat Wave Dynamics and Predictability over the Eastern Mediterranean"

_Earth System Dynamics, 2020_

## Referee Comment (RC1) · Anonymous Referee #1 · 12 Aug 2020

**Review of Hochman *et al.*, 2020**

**A New View of Heat Wave Dynamics and Predictability over the Eastern Mediterranean**

In this paper, the authors employ an approach from dynamical systems theory to quantify the (intrinsic) predictability of atmospheric states based on reanalysis data during cold and hot extremes over the Eastern Mediterranean. This is complemented with GEFS reforecasts, which are used to infer forecast uncertainty, or practical predictability. While the distinction and investigation of practical and intrinsic predictability is not new (e.g. Melhauser & Zhang, 2012), I am not aware of any comparable publications in the context of heatwaves. In addition, a simple Lagrangian model is used to reveal the origin of near-surface air masses during hot and cold extreme events. The resulting paper is nicely structured, not too lengthy and certainly an interesting read. I only have two minor comments and a few additional comments, questions and suggestions, as the manuscript is well written and understandable.

- In Melhauser & Zhang (2012), the classic Lorenz (1969) paper is cited multiple times; whereas intrinsic predictability is first defined as "the extent to which prediction is possible if an optimum procedure is used", it is then also related to knowledge of the 'atmospheric state'.
  Now, here, the authors do not cite any study when they claim that "As opposed to the practical predictability, the intrinsic predictability only depends on the characteristics of the atmosphere itself." Even though I understand that authors might want to stick to historical definitions, to me, it makes only little sense to limit the forecasting system to the atmosphere. An increasing amount of evidence shows that the Earth's surface does not only supply the atmosphere with heat and moisture, but, to some extent, also exerts control over it (e.g. Koster *et al.*, 2010; Dirmeyer *et al*., 2018). Knowledge of the land (and ocean) surface state thus implies improved predictability up to sub-seasonal timescales (thanks to, e.g., soil memory).
  To be clear, I believe the focus on the atmosphere in this study makes sense, but I would still like to question why this 'intrinsic' predictability should be purely atmospheric by any means. To me, it seems like Lorenz (1969) emphasized the knowledge of all governing equations as well as observing the initial state, and I do not see why this would not include other components of the Earth System.
- Concerning the 2010 heatwave analysis: while it is interesting to show that the air parcels tracked back in time were warmer than on average even 10 days in the past in this specific case, Bieli *et al.* (2015), e.g., already found that high temperatures in the Balkans area tend to be primarily the result of high starting temperatures combined with extensive descent, enabling strong adiabatic heating.
  The authors also attempt to explain why particularly the metrics calculated on SLP differ so strongly from those calculated on Z500. I am using this opportunity to refer to my previous comment here – perhaps the fact that the evolution of the atmospheric

state closer to the surface tends to follow less of a clear pattern, or "the larger spread in dynamical systems properties across the different heat waves for SLP than for Z500", is partially caused by interactions with the land surface. Naturally, these interactions predominantly affect the lowermost parts of the troposphere, and to provide an example, it can actually be seen (if the different units are accounted for) in Fig. 12 of Zampieri *et al.* (2009) that unusually dry soils affect SLP more than Z500 in a modelling experiment.

I thus think that the 2010 heatwave part needs a bit more attention, as currently, the main message is that the single case is similar to the climatology with respect to Z500 evolution during heatwaves, but highly different in terms of SLP, and in my opinion, this is not explained sufficiently (see also comments above and concerning L. 296 below).

**Further comments**
- L. 36: Feeling the heat**,** 2018 (comma missing?)
- L. 49: Saaroni and Ziv,2000) (space missing?)
- L. 126: If the CSI comprises the boundary layer height, then why is it absent from the equation below (L. 130)? Also, in the cited Saaroni *et al.* (2017), the atmospheric boundary layer height itself is barely mentioned, but rather the (height of the) persistent marine inversion. I thus recommend slightly editing (or shortening) this part to further enhance the consistency and clarity of the text.
- L. 212: Is there any reason for this choice, i.e. initializing trajectories between the surface and 90 hPa above, other than simplicity? To me, it seems more intuitive to always track the air masses within the atmospheric boundary layer back in time, whose height may vary from day to day, and tends to be (positively) anomalous particularly during heatwaves (this might not be the case in the study area of interest, but was certainly true for the 'epicenter' of the 2010 Russian heatwave; see, e.g. Miralles *et al.*, 2014). However, considering that Dayan *et al.* (2002) demonstrated only little synoptic-scale influence on summer ABL heights compared to the distance to the coast, the usage of a constant layer to be tracked backward might be entirely justified, but perhaps the authors can still elaborate on their choice.
- L. 240: It is quite interesting that the specific humidity does not increase nearly as much in the last 48 hours prior to arrival during heatwaves as for cold extremes, but is this only a consequence of different 'inflow', i.e. more trajectories over the Mediterranean sea? Of course, this cannot be gauged solely by a visual comparison of Figs. 2a & 2b (in which, to me, the trajectory densities shortly before arrival do not seem to differ much), but I would also suspect that additional factors are at play – such as, e.g., enhanced convective activity (and hence moistening of the troposphere).
- L. 264: "The build-up towards this type of event is characterized by an increase in $\theta$ (decrease in persistence) and a decrease in d (Fig. 4d)." This comment also concerns the Methods section; I think the authors provide a good overview of the two dynamical system metrics, but perhaps it would be helpful to explicitly state that, as explained by Moloney *et al*. (2019), more (expanding) dimensions around the state of interest (or degrees of freedom, I suppose) imply less predictability. Or, in other words, lower d

suggests higher predictability – this is actually stated as such on L. 335 in the Summary, but as far as am I concerned, not before. Perhaps it would seem a bit confusing to edit the sentence I am quoting above (L. 264), as in this example (Fig. 4d), persistence decreases, yet predictability as gauged by the local dimension d increases. Still, I believe I am not the only reader who would appreciate a bit more guidance throughout the manuscript.

- L. 280: "The pattern somewhat resembles the temporal evolution of d computed on SLP (cf. Fig. 5e and Fig. 4c), but stands in stark contrast to the pattern computed on Z500". While I agree that there is a stark contrast to the pattern shown in Fig. 4a (Z500), I find this resemblance a bit difficult to spot, as the peaks in d and msl spread appear to be shifted by about one day. Is there any obvious reason for this? Also, I imagine this would look different for shorter or longer lead times, so come to think of it, why not 24 hours less or more? This is not a request to repeat the entire analysis for different lead times (probably out of scope anyways), but I am just curious if the authors looked into this and if so, how much this choice even matters in the first place.

- L. 296: " We further hypothesize that differences between the single case and the climatology may be related to the relatively small day to day variations during summer over the Eastern Mediterranean (Ziv *et al*., 2004), which make it challenging to depict the exact onset of a heat wave". Could you please elaborate how the challenges related to defining the onset of a heatwave could contribute to the extreme differences between the 2010 case and the climatology, but only for SLP and not Z500 – this is not obvious to me.

- L. 303: Concerning the anticyclonic wave breaking, if my understanding is correct, then this can be seen in Fig. 8, as the trough east of the ridge centered over European Russia, clearly visible from Fig. 8b onward, is tilted (southwest-northeast; Davini *et al*., 2012) and advected westward (consistent with the definition given in Quandt *et al*., 2019). I suggest adding a brief description along these lines for readers unfamiliar with the terminology, this would also prevent readers from overlooking Fig. 8 (which, currently, is only mentioned but not discussed in the main text).

- L. 305: (Quandt *et al*., 2019)**,** played (comma missing?)

- L. 322 (+ 365, 511): K**ue**ne ⇒ Keune, typo.

- Fig. 5: While a few sentences in the Methods explain what is really shown in Fig. 5, I believe the caption might benefit from a small addition, hinting at the fact that results are plotted for their corresponding initialisation times.

- Fig. 9: I suggest using red colors for the upper 10% of CSI, as in previous figures, and plotting the 2010 heatwave results in black (or any other color than blue) instead.

**References**

Bieli, M., Pfahl, S. & Wernli, H. A lagrangian investigation of hot and cold temperature extremes in Europe. Q. J. R. Meteorol. Soc. 141, 98–108 (2015).

Davini, P., Cagnazzo, C., Gualdi, S. & Navarra, A. Bidimensional diagnostics, variability, and trends of northern hemisphere blocking. J. Clim. 25, 6496–6509 (2012).

Dayan, U., Lifshitz-Goldreich, B. & Pick, K. Spatial and structural variation of the atmospheric boundary layer during summer in Israel-profiler and rawinsonde measurements. J. Appl. Meteorol. 41, 447–457 (2002).

Dirmeyer, P. A., Halder, S. & Bombardi, R. On the Harvest of Predictability From Land States in a Global Forecast Model. J. Geophys. Res. Atmos. 123, 13,111-13,127 (2018).

Koster, R. D. *et al.* Contribution of land surface initialization to subseasonal forecast skill: First results from a multi-model experiment. Geophys. Res. Lett. 37, 1–6 (2010).

Levi, Y., Shilo, E. & Setter, I. Climatology of a summer coastal boundary layer with 1290-MHz wind profiler radar and a WRF simulation. J. Appl. Meteorol. Climatol. 50, 1815–1826 (2011).

Lorenz, E. N. Atmospheric Predictability as Revealed by Naturally Occurring Analogues. J. Atmos. Sci. 26, 636–646 (1969).

Miralles, D. G., Teuling, A. J., Van Heerwaarden, C. C. & De Arellano, J. V. G. Mega-heatwave temperatures due to combined soil desiccation and atmospheric heat accumulation. Nat. Geosci. 7, 345–349 (2014).

Melhauser, C. & Zhang, F. Practical and intrinsic predictability of severe and convective weather at the mesoscales. J. Atmos. Sci. 69, 3350–3371 (2012).

Moloney, N. R., Faranda, D. & Sato, Y. An overview of the extremal index. Chaos 29, (2019).

Quandt, L. A., Keller, J. H., Martius, O., Pinto, J. G. & Jones, S. C. Ensemble sensitivity analysis of the blocking system over Russia in summer 2010. Mon. Weather Rev. 147, 657–675 (2019).

Saaroni, H., Savir, A. & Ziv, B. Synoptic classification of the summer season for the Levant using an 'environment to climate' approach. Int. J. Climatol. 37, 4684–4699 (2017).

Zampieri, M. *et al.* Hot European Summers and the Role of Soil Moisture in the Propagation of Mediterranean Drought. J. Clim. 22, 4747–4758 (2009).

---

## Referee Comment (RC2) · Anonymous Referee #2 · 14 Oct 2020

Review of: A New View of Heat Wave Dynamics and Predictability over the Eastern Mediterranean Author(s): Assaf Hochman et al. MS No.: esd-2020-37 MS Type: Research article

General comments:

The study titled "A New View of Heat Wave Dynamics and Predictability over the Eastern Mediterranean" by Hochman et al., presents a fresh viewpoint on the dynamics and predictability of heat waves in the Eastern Mediterranean by using both dynamical system theory and an ensemble of NWP models. This study continues the line of previous papers by the authors, dealing with the dynamics of cold spells, and weather regimes in the Eastern Mediterranean in general, which is an interesting viewpoint on a well-studied subject. This study stands out, by showing comparisons to ensemble

of NWP results, which seems really promising and puts this viewpoint in perspective. The study presents the climatology of the heat waves from this viewpoint, along with a complementary analysis of back-trajectories, showing the origins of air parcels – which is interesting on its own. Furthermore, a specific heat-wave case-study is presented and analyzed, showing the evolution of this unique heat wave.

To my view, the study addresses relevant and current problems, and shows a novel concept to answering such problems. I think this paper is worth publishing in ESD, after addressing some points, as follows:

Sect. 2.3, and in general: I feel that there is not enough "intuition" in the description of the dynamical system metrics. For example, it is hard for me to understand what exactly is the meaning of ðİIJČ−1 ("quantifies the persistence of the system in the neighborhood of the state of interest, and tends to be very sensitive to small changes in the state of the system"). I do realize this notion was already mentioned in quite a few papers in recent years, however, I feel that me (and the other readers) of this paper can benefit from a more intuitive explanation of the metrics. Moreover, the possibility of other readers to repeat such an analysis is limited by the fact you refer to other studies on how to estimate the parameters. I believe the parameter estimation should be further detailed, at least in the supplementary material, including the report on the errors in estimating the parameters.

It could be worthy to expand much the discussion about the differences between the intrinsic and the practical predictability. Such a discussion could be exampled and explained using the presented case study, by showing how exactly can you elevate the dynamical system theory in predicting this heatwave better than using only the ensemble of NWP. Could this be performed and displayed in the paper?

Another point which I think you should address, is the portion of the back-trajectories which is terrestrial vs. the marine portion. It seems to me there could be much of a difference between the heat waves and the cold days. Am I correct?

Specific comments

L44: "(e.g., Goldreich et al., 2003)". Consider citing (Kushnir et al., 2017).

L48-L52: some of this description seems appropriate to describe the generally mild temperatures and small inter-daily variability which was mentioned at the beginning of the paragraph. Please consider having some of this description moved to the beginning of the paragraph, and leave out only the part which is unique to low-temp days (possibly emphasize the role of the upper-level trough).

L58: "∼55,000 excess deaths": where?

L62: "record-breaking": in terms of duration? Extent? Temperatures? Please address this in the text.

L69: "A framework": which framework? Is it yours or in general?

L133: The term "Etesian winds" was not introduced before. To me it seems like a good idea to present it in the paragraph describing the summer climatology of the Eastern Mediterranean.

Sect. 2.2: please elaborate on why the CSI is better at describing heat waves than, e.g., the temperature alone. This could be done by using examples or just a further explanation on other effects these heat waves consist of.

L174-179: Does this seasonal cycle related to the synoptic-scale circulation? If it is, I am not sure why is it reasonable to subtract it from the data.

L190: Could you write explicitly if this interpolation is done on the horizontal axis only or on the vertical axis as well?

L191: I did not understand what the reason was for choosing 69 hours as the lead time. Please elaborate.

L200: the bootstrapping and the statistical tests are already mentioned elsewhere,

and to my opinion should not be detailed twice. However, it will be good if you could explain what was the variable on which the bootstrapping was applied on, and how many repetitions were made.

L209-210: where do you start the trajectories from? Are they spread allover the domain? Is it only from the 5 stations?

L224: "upper level ridge" vs. trough: please write more precisely, that the center of the high is to the southeast of the study area, as it is hard to tell from this map weather the Eastern Mediterranean is affected by the ridge or the trough to the northwest (it actually seems in between them).

L224-226: please make sure the SLP intervals are the same in panels a and b, as it is hard to understand which of these situations is associated with a deeper/shallower longer/shorter Persian trough (might be worth mentioning this as well).

L227-242: I am not convinced that the median back-trajectory is a good representation of the paths of the air parcels. For example, the median track for the cool days is out of the highest density region (if I understand the plot correctly). This means, it could also be some compromise between trajectories passing over the Black Sea, and trajectories passing over the Mediterranean. Could you please explain why the median is a good representation? Could you convince me (and the readers) why should those maps not be read differently? For example, one can argue the main difference between the trajectories is that during heat waves more tracks are arriving after the passage over continental regions (Turkey), while during cold spells, tracks are arriving from the Black and Aegean seas.

L251: "Zero..." please add one of the following, or a similar description: x-axis / Time / abscissa.

L267-275: Could you please give an intuition about the numbers shown in Fig. 4? For example, what does zero on the y-axis means? What is the difference between an

increase of d and an increase of theta?

L277-283: It is not clear to me what can we learn from the abserr graphs. Is this the error computed relative to the stations?

L306: What do we learn from Fig. 8? Please enhance its description or cut it out of the main body (it could be transferred to the supplementary).

L390: Could you also provide a table showing the d and Theta for the analyzed times?

Figure 1: The colorscale of panels a and b is not the same (panel a uses green colors in the middle of the SLP range, while panel b uses only yellows). Please match the colorscales. Furthermore, please either write the interval of the SLP contours or add labels to some of them, so it could be easier to compare between the plots.

Figure 3: either the SLP and Z500 labels were swapped or their mentioning in the figure caption. Please also explain what is represented by each dot. Is it the 12 UTC d and theta from the NCEP for each of the analyzed days? If so, please write something in that spirit.

Figure 7a,b: Could you please make the blue colors somewhat transparent? It is harder to read the map in the opaque form of the trajectories densities.

Figure S1: Please add either topography or some measure of the summer-climate (average temp. / max temp.) to the map. In this way the readers could assess why does the 5 stations are representative of the climate.

Technical corrections

L46: "On the upper levels": please consider adding the words "of the troposphere".

L49: "Saaroni and Ziv,2000": please add a space before the "2000".

L89: Please consider deleting the word "thus".

L141: "...nine out of eleven days": please add "on average".

References

Kushnir, Y., Dayan, U., Ziv, B., Morin, E. and Enzel, Y.: Climate of the Levant: phenomena and mechanisms, in Quaternary of the Levant: environments, climate change, and humans, edited by Y. Enzel and B.-Y. Ofer, pp. 31–44, Cambridge University Press, Cambridge, UK., 2017.

---

## Author Comment (AC1) · 4 Nov 2020

esd-2020-37: "A new view of heat wave dynamics and predictability over the Eastern Mediterranean" by Assaf Hochman, Sebastian Scher, Julian Quinting, Joaquim Pinto and Gabriele Messori.

**Point by point response to Reviewer 1:**

**Reviewer 1:** In this paper, the authors employ an approach from dynamical systems theory to quantify the (intrinsic) predictability of atmospheric states based on reanalysis data during cold and hot extremes over the Eastern Mediterranean. This is complemented with GEFS reforecasts, which are used to infer forecast uncertainty, or practical predictability. While the distinction and investigation of practical and intrinsic predictability is not new (e.g. Melhauser & Zhang, 2012), I am not aware of any comparable publications in the context of heatwaves. In addition, a simple Lagrangian model is used to reveal the origin of near-surface air masses during hot and cold extreme events. The resulting paper is nicely structured, not too lengthy and certainly an interesting read. I only have two minor comments and a few additional comments, questions and suggestions, as the manuscript is well written and understandable.

**Response:** Thank you for the positive feedback. We plan on addressing all of the Reviewer's comments in the revised version of the manuscript as described below.

**Reviewer 1:** In Melhauser & Zhang (2012), the classic Lorenz (1969) paper is cited multiple times; whereas intrinsic predictability is first defined as "the extent to which prediction is possible if an optimum procedure is used", it is then also related to knowledge of the 'atmospheric state'.

Now, here, the authors do not cite any study when they claim that "As opposed to the practical predictability, the intrinsic predictability only depends on the characteristics of the atmosphere itself." Even though I understand that authors might want to stick to historical definitions, to me, it makes only little sense to limit the forecasting system to the atmosphere. An increasing amount of evidence shows that the Earth's surface does not only supply the atmosphere with heat and moisture, but, to some extent, also exerts control over it (e.g. Koster *et al.*, 2010; Dirmeyer *et al.*, 2018). Knowledge of the land (and ocean) surface state thus implies improved predictability up to sub-seasonal timescales (thanks to, e.g., soil memory).

To be clear, I believe the focus on the atmosphere in this study makes sense, but I would still like to question why this 'intrinsic' predictability should be purely atmospheric by any means. To me, it seems like Lorenz (1969) emphasized the knowledge of all governing equations as well as observing the initial state, and I do not see why this would not include

other components of the Earth System.

**Response:** We agree with the Reviewer's comment. Indeed, the atmospheric state depends not only on the atmosphere, but is influenced by interactions with land and ocean. However, it is important to note that in many – albeit certainly not all – cases these interactions influence the atmosphere at time scales longer than those we consider in our analysis (e.g., Entin *et al*., 2000), and act as a seasonal-scale preconditioning to extremely high summer temperatures (e.g. the mechanism discussed in the Zampieri *et al*. (2009) study the Reviewer cites in his second comment). We plan on rephrasing the sentences related to this notion. We will specifically clarify that, while we take a predominantly atmospheric perspective, centered on synoptic timescales, there is ample evidence of the importance of surface interactions with other "spheres" of the climate system, not least for controlling atmospheric predictability (such as in the experiments described in the Koster *et al*. (2010) study the Reviewer pointed us to). In this respect, we will read and reference the relevant literature suggested by the Reviewer.

**Reference:**

Entin JK, Robock A, Vinnikov KY, Hollinger SE, Liu S, Namkhai A. 2000. Temporal and spatial scales of observed soil moisture variations in the extra tropics. *Journal of Geophysical Research* **105(D9)**: 11865– 11877. https://doi.org/10.1029/2000JD900051

**Reviewer 1:** Concerning the 2010 heatwave analysis: while it is interesting to show that the air parcels tracked back in time were warmer than on average even 10 days in the past in this specific case, Bieli *et al.* (2015), e.g., already found that high temperatures in the Balkans area tend to be primarily the result of high starting temperatures combined with extensive descent, enabling strong adiabatic heating. The authors also attempt to explain why particularly the metrics calculated on SLP differ so strongly from those calculated on Z500. I am using this opportunity to refer to my previous comment here – perhaps the fact that the evolution of the atmospheric state closer to the surface tends to follow less of a clear pattern, or "the larger spread in dynamical systems properties across the different heat waves for SLP than for Z500", is partially caused by interactions with the land surface. Naturally, these interactions predominantly affect the lowermost parts of the troposphere, and to provide an example, it can actually be seen (if the different units are accounted for) in Fig. 12 of Zampieri *et al.* (2009) that unusually dry soils affect SLP more than Z500 in a modelling experiment. I thus think that the 2010 heatwave part needs a bit more attention, as currently, the main message is that the

single case is similar to the climatology with respect to Z500 evolution during heatwaves, but highly different in terms of SLP, and in my opinion, this is not explained sufficiently (see also comments above and concerning L. 296 below).

**Response:** Thank you for this comment and for pointing us to some useful references we had overlooked. We agree that the discussion of the 2010 heat wave could be extended with respect to the differences between the dynamical systems analysis close to the surface and at 500 hPa. We specifically plan to discuss the possible influence air-sea interactions may have on the dynamical systems metrics at surface level. Soil moisture, on the other hand, may not be an important factor for controlling heat waves over the South-Eastern part of the Mediterranean, but rather more important in the North-Eastern areas (Zittis *et al*., 2014). It may however be possible that low soil moisture in the regions where the air parcels originate influence the intensity of Eastern Mediterranean heat waves. We will add a discussion of this point. We will further read the references suggested by the Reviewer and refer to them in the revised version of the manuscript.

**Reference:**

Zittis G, Hadjinicolaou P, Lelieveld J. 2014. Role of soil moisture in the amplification of climate warming in the eastern Mediterranean and the Middle East. *Climate Research* 59: 27-37. https://doi.org/10.3354/cr01205

**Reviewer 1:** L. 36: Feeling the heat**,** 2018 (commamissing?)
**Response:** A comma will be added.

**Reviewer 1:** L. 49: Saaroni and Ziv,2000 (spacemissing?)
**Response:** A space will be added.

**Reviewer 1:** L. 126: If the CSI comprises the boundary layer height, then why is it absent from the equation below (L. 130)? Also, in the cited Saaroni *et al.* (2017), the atmospheric boundary layer height itself is barely mentioned, but rather the (height of the) persistent marine inversion. I thus recommend slightly editing (or shortening) this part to further enhance the consistency and clarity of the text.
**Response:** We agree with the Reviewer and will reformulate this part of the manuscript according to Saaroni et al. (2017).

**Reviewer 1:** L. 212: Is there any reason for this choice, i.e. initializing trajectories between the surface and 90 hPa above, other than simplicity? To me, it seems more intuitive to always track the air masses within the atmospheric boundary layer back in time, whose height may vary from day to day, and tends to be (positively) anomalous particularly during heatwaves (this might not be the case in the study area of interest, but was certainly true for the 'epicenter' of the 2010 Russian heatwave; see, e.g. Miralles *et al.*, 2014). However, considering that Dayan *et al.* (2002) demonstrated only little synoptic-scale influence on summer ABL heights compared to the distance to the coast, the usage of a constant layer to be tracked backward might be entirely justified, but perhaps the authors can still elaborate on their choice.

**Response:** Thank you for this comment. According to recent studies, the planetary boundary layer height in Israel during summer is ~600 – 900m above the surface (Uzan *et al.*, 2016; 2020). Assuming that the pressure decreases by 1 hPa every 8m height difference, 90 hPa corresponds to about 720m, thus 90hPa can be considered a reasonable choice.

We have further evaluated the trajectories that start from 90hPa above the surface and higher (Figure R1). We find that at these levels the picture is somewhat different. Air parcels associated with heat waves are located over the Mediterranean Sea and east of Israel prior to the heat wave (Figure R1a). Air parcels associated with cool days originate mostly over the Atlantic and Europe and are transported downstream (Figure R1b). This further indicates that our trajectory analysis successfully identifies air parcels in the boundary layer, which have a different path than those in the free troposphere. Moreover, the thermodynamic properties of the air parcels do not show any noticeable differences between heat waves and cool days (Figure R1c-f). We will elaborate on the choices we made with regards to the back trajectories' analysis as described above in the revised version of the manuscript.

**Reference**

Uzan L, Egert S, Alpert P. 2016. Ceilometer evaluation of the eastern Mediterranean summer boundary layer height – first study of two Israeli sites. *Atmospheric Measurement Techniques* **9**: 4387–4398. https://doi.org/10.5194/amt-9-4387-2016

Uzan L, Egert S, Khain P, Levi Y, Vadislavsky E, Alpert P. 2020. Ceilometers as planetary boundary layer height detectors and a corrective tool for COSMO and IFS models. *Atmospheric Chemistry and Physics* 20: 12177-12192. https://doi.org/10.5194/acp-20-12177-2020

[Figure]

**Figure R1** Same as Figure 2, but for trajectories initialized from 90hPa above the surface and higher with the uncertainty around the median trajectories of 25 – 75% of the trajectories.

**Reviewer 1:** L. 240: It is quite interesting that the specific humidity does not increase nearly as much in the last 48 hours prior to arrival during heatwaves as for cold extremes, but is this only a consequence of different 'inflow', i.e. more trajectories over the Mediterranean Sea? Of course, this cannot be gauged solely by a visual comparison of Figs. 2a & 2b (in which, to me, the trajectory densities shortly before arrival do not seem to differ much), but I would also suspect that additional factors are at play – such as, e.g., enhanced convective activity (and hence moistening of the troposphere).

**Response:** Thank you very much for this suggestion. Our additional analysis reveals that for most of the time, the portion of terrestrial back-trajectories is similar for heat waves and cool days (Figure R2). It is only at about 72 to 24 hours prior to the events that the portion of terrestrial (marine) back-trajectories is lower (higher) for cool days than for heat waves (Figure R2). This is in line with the evolution in specific humidity along the trajectories, which increases more strongly for the cool days than for the heat waves (Fig. 2f). The increase of moisture is most likely related to the passing of the air masses over the Mediterranean Sea. As a caveat, this analysis does not account for local processes such as the convective activity mentioned by the Reviewer. We will revise the text accordingly and consider adding Figure R2 as a Supplementary Figure.

[Figure]

**Figure R2** The portion of trajectories over land for heat waves (red line) and cool days (blue line).

**Reviewer 1:** L. 264: "The build-up towards this type of event is characterized by an increase in θ (decrease in persistence) and a decrease in d (Fig. 4d)." This comment also concerns the Methods section; I think the authors provide a good overview of the two dynamical system metrics, but perhaps it would be helpful to explicitly state that, as explained by Moloney *et al.* (2019), more (expanding) dimensions around the state of interest (or degrees of freedom, I suppose) imply less predictability. Or, in other words, lower d suggests higher predictability – this is actually stated as such on L. 335 in the Summary, but as far as am I concerned, not before. Perhaps it would seem a bit confusing to edit the sentence I am quoting above (L. 264), as in this example (Fig. 4d), persistence decreases, yet predictability as gauged by the local dimension d increases. Still, I believe I am not the only reader who would appreciate a bit more guidance throughout the manuscript.

**Response:** We will clarify these points and provide a more intuitive qualitative interpretation of the metrics earlier in the manuscript, focusing on their relation with the intrinsic predictability. To that end, we have completely re-structured Sect. 2.3 to provide both a

clearer intuitive explanation of the metrics and a more detailed description of the mathematical background to ensure the reproducibility of our results.

**Reviewer 1:** L. 280: "The pattern somewhat resembles the temporal evolution of d computed on SLP (cf. Fig. 5e and Fig. 4c), but stands in stark contrast to the pattern computed on Z500". While I agree that there is a stark contrast to the pattern shown in Fig. 4a (Z500), I find this resemblance a bit difficult to spot, as the peaks in d and msl spread appear to be shifted by about one day. Is there any obvious reason for this? Also, I imagine this would look different for shorter or longer lead times, so come to think of it, why not 24 hours less or more? This is not a request to repeat the entire analysis for different lead times (probably out of scope anyways), but I am just curious if the authors looked into this and if so, how much this choice even matters in the first place.

**Response:** Thank you for this comment. Since the spread and error are computed every 24 hours and the dynamical systems metrics are instantaneous in time (local in phase-space) and computed from the 6-hourly data, we believe that a shift of up to 24-hours may be reasonable. We have further tested a 24-hour shorter lead time and found that the peak in the spread is at about 0 h, resembling the peak in *d* computed on SLP (Figure R3, left panel). We will discuss this point briefly in the revised text, as we have indeed not described in detail the sensitivity of our results to the choice of lead-time.

[Figure]

**Figure R3** Same as Figure 5e, f but for a 24-hour shorter lead time.

**Reviewer 1:** L. 296: "We further hypothesize that differences between the single case and the climatology may be related to the relatively small day to day variations during summer over the Eastern Mediterranean (Ziv *et al*., 2004), which make it challenging to depict the exact onset of a heat wave". Could you please elaborate how the challenges related to defining the onset of a heatwave could contribute to the extreme differences between the 2010 case and the climatology, but only for SLP and not Z500 – this is not obvious to me.

**Response:** When comparing the climatology of the temporal evolution of *d* and *θ* for Z500

(Fig. 4a) with the single case (Fig. 6b) it is relatively easy to see that in both there is an increase in both $d$ and $\theta$ as the heat wave develops. On the other hand, when comparing the temporal evolution of $d$ and $\theta$ for SLP (Fig. 4c with Fig. 6c), one can see that depicting the exact time the heat wave starts is very important for comparison. Still, in both Figures $d$ increases and $\theta$ decreases at some point in the chosen time window, but the timing of these trends is shifted between the climatology (Fig. 4c) and the single case (Fig. 6c). We will clarify this in the revised version of the manuscript.

**Reviewer 1:** L. 303: Concerning the anticyclonic wave breaking, if my understanding is correct, then this can be seen in Fig. 8, as the trough east of the ridge centered over European Russia, clearly visible from Fig. 8b onward, is tilted (southwest-northeast; Davini *et al.*, 2012) and advected westward (consistent with the definition given in Quandt *et al.*, 2019). I suggest adding a brief description along these lines for readers unfamiliar with the terminology, this would also prevent readers from overlooking Fig. 8 (which, currently, is only mentioned but not discussed in the main text).

**Response:** Thank you for this suggestion. The Reviewer is correct with regard to the anticyclonic Rossby wave breaking shown in Fig. 8. We will add some more information on this for the reader and discuss Fig. 8 more in detail in the revised version of the manuscript.

**Reviewer 1:** L. 305: (Quandt *et al.*, 2019)**,** played (comma missing?)
**Response:** A comma will be added.

**Reviewer 1:** L. 322 (+ 365, 511): K**ue**ne ⇒ Keune
**Response:** The typo will be corrected in all mentioned lines.

**Reviewer 1:** Fig. 5: While a few sentences in the Methods explain what is really shown in Fig. 5, I believe the caption might benefit from a small addition, hinting at the fact that results are plotted for their corresponding initialization times.
**Response:** We will expand the caption of Fig. 5 to explain that the results are plotted for initialization time.

**Reviewer 1:** Fig. 9: I suggest using red colors for the upper 10% of CSI, as in previous figures, and plotting the 2010 heatwave results in black (or any other color than blue) instead.
**Response:** We will change the Figure according to the Reviewer's suggestion (see below new

Figure 9).

[Figure]

**New Figure 9** Forecast spread/skill for the mid-August heat wave centered (0 h) on 15.8.2010 at 12UTC (black line). The mean temporal evolution of the ensemble model spread for Tmax (a), SLP (c) and absolute error for Tmax (b) of forecasts with lead-time 69h, initialized at different time lags with respect to the event, computed every 24 hours. The heat waves (upper 10% of CSI - red line) are displayed for reference. A 95% bootstrap confidence interval for all heatwaves is displayed in shading.

**Reviewer 1:** References

Bieli, M., Pfahl, S. & Wernli, H. A lagrangian investigation of hot and cold temperature extremes in Europe. Q. J. R. Meteorol. Soc. 141, 98–108 (2015).

Davini, P., Cagnazzo, C., Gualdi, S. & Navarra, A. Bidimensional diagnostics, variability, and trends of northern hemisphere blocking. J. Clim. 25, 6496–6509 (2012).

Dayan, U., Lifshitz-Goldreich, B. & Pick, K. Spatial and structural variation of the atmospheric boundary layer during summer in Israel-profiler and rawinsonde measurements.

J. Appl. Meteorol. 41, 447–457 (2002).

Dirmeyer, P. A., Halder, S. & Bombardi, R. On the Harvest of Predictability From Land States in a Global Forecast Model. J. Geophys. Res. Atmos. 123, 13,111-13,127 (2018).

Koster, R. D. *et al.* Contribution of land surface initialization to subseasonal forecast skill: First results from a multi-model experiment. Geophys. Res. Lett. 37, 1–6 (2010).

Levi, Y., Shilo, E. & Setter, I. Climatology of a summer coastal boundary layer with 1290-MHz wind profiler radar and a WRF simulation. J. Appl. Meteorol. Climatol. 50, 1815–1826 (2011).

Lorenz, E. N. Atmospheric Predictability as Revealed by Naturally Occurring Analogues. J. Atmos. Sci. 26, 636–646 (1969).

Miralles, D. G., Teuling, A. J., Van Heerwaarden, C. C. & De Arellano, J. V. G. Mega-heatwave temperatures due to combined soil desiccation and atmospheric heat accumulation. Nat. Geosci. 7, 345–349 (2014).

Melhauser, C. & Zhang, F. Practical and intrinsic predictability of severe and convective weather at the mesoscales. J. Atmos. Sci. 69, 3350–3371 (2012).

Moloney, N. R., Faranda, D. & Sato, Y. An overview of the extremal index. Chaos 29, (2019).

Quandt, L. A., Keller, J. H., Martius, O., Pinto, J. G. & Jones, S. C. Ensemble sensitivity analysis of the blocking system over Russia in summer 2010. Mon. Weather Rev. 147, 657–675 (2019).

Saaroni, H., Savir, A. & Ziv, B. Synoptic classification of the summer season for the Levant using an 'environment to climate' approach. Int. J. Climatol. 37, 4684–4699 (2017).

Zampieri, M. *et al.* Hot European Summers and the Role of Soil Moisture in the Propagation of Mediterranean Drought. J. Clim. 22, 4747–4758 (2009)

**Response:** Thank you for providing these references. We plan on citing them where applicable and referring to them in the revised text.

---

## Author Comment (AC2) · 4 Nov 2020

esd-2020-37: "A new view of heat wave dynamics and predictability over the Eastern Mediterranean" by Assaf Hochman, Sebastian Scher, Julian Quinting, Joaquim Pinto and Gabriele Messori.

**Point by point response to Reviewer 2:**

**Reviewer 2**: The study titled "A New View of Heat Wave Dynamics and Predictability over the Eastern Mediterranean" by Hochman et al., presents a fresh viewpoint on the dynamics and predictability of heat waves in the Eastern Mediterranean by using both dynamical system theory and an ensemble of NWP models. This study continues the line of previous papers by the authors, dealing with the dynamics of cold spells, and weather regimes in the Eastern Mediterranean in general, which is an interesting viewpoint on a well-studied subject. This study stands out, by showing comparisons to ensemble NWP results, which seems really promising and puts this viewpoint in perspective. The study presents the climatology of the heat waves from this viewpoint, along with a complementary analysis of back-trajectories, showing the origins of air parcels – which is interesting on its own. Furthermore, a specific heat-wave case-study is presented and analyzed, showing the evolution of this unique heat wave. To my view, the study addresses relevant and current problems, and shows a novel concept to answering such problems. I think this paper is worth publishing in ESD, after addressing some points, as follows:

**Response:** Thank you for the positive feedback. We plan on addressing all of the Reviewer's comments in the revised version of the manuscript as described below.

**Reviewer 2:** Sect. 2.3, and in general: I feel that there is not enough "intuition" in the description of the dynamical system metrics. For example, it is hard for me to understand what exactly is the meaning of $\theta^{-1}$ ("quantifies the persistence of the system in the neighborhood of the state of interest, and tends to be very sensitive to small changes in the state of the system"). I do realize this notion was already mentioned in quite a few papers in recent years, however, I feel that me (and the other readers) of this paper can benefit from a more intuitive explanation of the metrics. Moreover, the possibility of other readers to repeat such an analysis is limited by the fact you refer to other studies on how to estimate the parameters. I believe the parameter estimation should be further detailed, at least in the supplementary material, including the report on the errors in estimating the parameters.

**Response:** Thank you for this comment. We understand the need for both a clearer intuitive

explanation, to aid the understanding of our study, and a more complete analytical derivation of the metrics, to ensure that our study is self-contained and allows reproducibility of our results. To this end, we have completely re-structured Sect. 2.3 to provide both a clearer intuitive explanation of the metrics and a more detailed description of the mathematical background. We also reference earlier uncertainty estimates for the calculation of the parameters.

**Reviewer 2:** It could be worthy to expand much the discussion about the differences between the intrinsic and the practical predictability. Such a discussion could be exampled and explained using the presented case study, by showing how exactly can you elevate the dynamical system theory in predicting this heatwave better than using only the ensemble of NWP. Could this be performed and displayed in the paper?

**Response:** Thank you for this important comment. The practical predictability relies on the performance of a numerical forecast model. As such, it blends model and data assimilation biases with the intrinsic characteristics of the atmospheric flow. Moreover, when restricting the analysis to a single realization of an ensemble forecast, such as in our case study, even a perfect ensemble may not provide a good skill-spread relationship. That is, even a perfect ensemble may have a spread that does not reflect the actual forecast error. In the specific case we analyze here, Tmax spread and error were well correlated, and matched an increase in local dimension. In other cases, the ensemble forecast can have, *posteriori*, a bad spread-error relationship. In these cases, local dimension and/or persistence trends that seem to contradict a low ensemble spread may serve as warning of a potentially poor spread-error relationship. We will extend the discussion on how exactly can the dynamical systems metrics be used to better predict heat waves, and support this by providing an example of a heatwave that displays the undesirable spread-error characteristics described above.

**Reviewer 2:** Another point which I think you should address, is the portion of the back-trajectories which is terrestrial vs. the marine portion. It seems to me there could be much of a difference between the heat waves and the cold days. Am I correct?

**Response:** Thank you very much for this suggestion. Our additional analysis reveals that for most of the time, the portion of terrestrial back-trajectories is similar for heat waves and cold days (Figure R1). It is only 72 to 24 hours prior to the events that the portion of terrestrial (marine) back-trajectories is lower (higher) for cool days than for heat waves (Figure R1). This is in line with the evolution in specific humidity along the trajectories, which increases more strongly for the cool days

than for the heat waves (Figure 2f). The increase of moisture is most likely related to the passing of the air masses over the Mediterranean Sea. We will revise the text accordingly and consider adding Figure R1 as a Supplementary Figure.

[Figure]

**Figure R1** The portion of trajectories over land for heat waves (red line) and cool days (blue line).

**Reviewer 2:** L44: "(e.g., Goldreich et al., 2003)". Consider citing (Kushnir et al., 2017).

**Response:** The reference will be cited in the revised version of the manuscript.

**Reviewer 2:** L48-L52: some of this description seems appropriate to describe the generally mild temperatures and small inter-daily variability which was mentioned at the beginning of the paragraph. Please consider having some of this description moved to the beginning of the paragraph, and leave out only the part which is unique to low-temp days (possibly emphasize the role of the upper-level trough).

**Response:** We will revise the paragraph according to the Reviewer's suggestion.

**Reviewer 2:** L58: "~55,000 excess deaths": where?

**Response:** In eastern Europe western Russia (Barriopedro et al., 2011; Katsafados et al., 2014). We will update the text accordingly.

**Reviewer 2:** L62: "record-breaking": in terms of duration? Extent? Temperatures? Please address this in the text.

**Response**: We will address this in the text. Indeed, record-breaking here is both in terms of temperatures and duration (https://ims.gov.il/sites/default/files/aug10.pdf ).

**Reviewer 2:** L69: "A framework": which framework? Is it yours or in general?

**Response:** The general framework of the Lagrangian back trajectories. We will clarify this in the text.

**Reviewer 2:** L133: The term "Etesian winds" was not introduced before. To me it seems like a good idea to present it in the paragraph describing the summer climatology of the Eastern Mediterranean.

**Response:** We appreciate this term may not be known to all readers, and we will introduce it in the paragraph describing the summer climatology of the eastern Mediterranean according to the Reviewer's suggestion.

**Reviewer 2:** Sect. 2.2: please elaborate on why the CSI is better at describing heat waves than, e.g., the temperature alone. This could be done by using examples or just a further explanation on other effects these heat waves consist of.

**Response:** We will elaborate on the advantages of the CSI index with respect to temperature alone. Indeed, this index incorporates the inversion base height and the heat stress. It therefore includes also humidity and circulation rather than only temperature (Saaroni *et al*., 2017). This means that the CSI relates more directly to the impacts of a heatwave on, for example, human physiology, than a conventional temperature-based measure (Epstein and Moran, 2006).

**Reference:**

Epstein Y, Moran DS. Thermal Comfort and the Heat Stress Indices. Vol. 44, Industrial Health. 2006.

**Reviewer 2:** L174-179: Does this seasonal cycle related to the synoptic-scale circulation? If it is, I am not sure why it is reasonable to subtract it from the data.

**Response:** We show that the seasonal cycle of the dynamical systems metrics is related to the synoptic scale circulation. Since we are comparing individual days/events during different parts of the summer season, it is better to de-seasonalize the data in order to study the anomalies. I.e., we test whether heat waves are synoptically and dynamically unusual with respect to the other days in the same part of the season. If we did not do this, the seasonal cycle would be "embedded" into our anomalies, biasing the results depending on which part of the summer season the selected episodes

occurred in. As a practical example, imagine a scenario where the seasonal cycle of *d* peaks during month "x" of summer, and is lowest during month "x+2". We next consider a heatwave during month "x" which has a *d* in-line with climatology, and a heatwave in month "x+2" which has a *d* above climatology for that month, but lower than the *d* of the heatwave during month "x". If we did not de-seasonalize *d* we would draw the incorrect conclusion that the first heatwave has an unusually high *d* and that the second heatwave has a low *d*. We will clarify this point in the revised version of the manuscript.

**Reviewer 2:** L190: Could you write explicitly if this interpolation is done on the horizontal axis only or on the vertical axis as well?

**Response:** We will clarify that the interpolation was done on the horizontal axis in the revised version of the manuscript.

**Reviewer 2:** L191: I did not understand what the reason was for choosing 69 hours as the lead time. Please elaborate.

**Response:** We will rephrase the paragraph in the methods section to clarify the choice of 69 hours. Indeed, The GEFS reforecasts are initialized at 00UTC and are available at three-hour intervals. Since our analysis focuses on heat waves, we estimate the spread/skill for maximum temperature and SLP at a lead-time of 69 hours, while the maximum temperature is defined between 45 h and 69 h. Given the three-hour interval of the forecast data, and bearing in mind that each station's maximum temperature is recorded between 20UTC and 20UTC of the next day, this time-window roughly corresponds to the definition of maximum temperature for the station data. We did however test a 24-hour shorter lead time and found that our main conclusions remain the same (see for example Figure R1).

[Figure]

**Figure R2** Same as Figure 5e, f but for a 24-hour shorter lead time.

**Reviewer 2:** L200: the bootstrapping and the statistical tests are already mentioned elsewhere, and to my opinion should not be detailed twice. However, it will be good if you could explain what was the variable on which the bootstrapping was applied on, and how many repetitions were made.

**Response**: Thank you for this suggestion. We will mention the statistical tests used only in Section 2.3 and elaborate that we bootstrapped the mean of the events and that the number of repetitions in the bootstrapping test is $10^4$.

**Reviewer 2**: L209-210: where do you start the trajectories from? Are they spread all over the domain? Is it only from the 5 stations?

**Response:** The trajectories are initialized from fixed points in the whole domain. We will specify this in the revised text.

**Reviewer 2:** L224: "upper level ridge" vs. trough: please write more precisely, that the center of the high is to the southeast of the study area, as it is hard to tell from this map weather the Eastern Mediterranean is affected by the ridge or the trough to the northwest (it actually seems in between them).

**Response:** We will clarify this point in the text.

**Reviewer 2:** L224-226: please make sure the SLP intervals are the same in panels a and b, as it is hard to understand which of these situations is associated with a deeper/shallower longer/shorter Persian trough (might be worth mentioning this as well).

**Response:** Thank you for this suggestion. We will change the Figure (see below new Figure 1) so that it will contain the same intervals in Fig. 1a and Fig. 1b according to the Reviewer's suggestion.

[Figure]

**New Figure 1** Mean sea level pressure (SLP in hPa, white contours) and 500 hPa geopotential height (Z500 in m, shaded in color) for the 10% of days with the highest (heat waves) and lowest (cool days) 'Climatic Stress Index' (CSI) values. (a) Heat wave days mean composite (b) Cool days mean composite (c) Heat waves minus cool days.

**Reviewer 2:** L227-242: I am not convinced that the median back-trajectory is a good representation of the paths of the air parcels. For example, the median track for the cool days is out of the highest density region (if I understand the plot correctly). This means, it could also be some compromise between trajectories passing over the Black Sea, and trajectories passing over the Mediterranean. Could you please explain why the median is a good representation? Could you convince me (and the readers) why should those maps not be read differently? For example, one can argue the main difference between the trajectories is that during heat waves more tracks are arriving after the passage over continental regions (Turkey), while during cold spells, tracks are arriving from the Black and Aegean seas.

**Response:** Thank you for this comment. We will revise Figure 2 so that it will contain the uncertainty around the median (New Figure 2). Indeed, we show that the uncertainty is relatively small in the - 48 hours prior to arrival of air parcels in the Eastern Mediterranean. This exemplifies that the median trajectory is a reasonable choice to make. As a caveat, this does not rule out a bimodal distribution of the trajectories, although the streamlines may be partly used to evaluate whether this is likely in the specific cases we consider here.

[Figure]

**New Figure 2** Same as Figure 2, but now also displaying the uncertainty of 25 – 75% around the median trajectories in crosses at different time lags.

**Reviewer 2:** L251: "Zero. . ." please add one of the following, or a similar description: x-axis / Time / abscissa.

**Response:** We will add one of the above in the revised version of the manuscript.

**Reviewer 2:** L267-275: Could you please give an intuition about the numbers shown in Fig. 4? For example, what does zero on the y-axis means? What is the difference between an increase of d and an increase of theta?

**Response:** The numbers shown in Fig. 4 are deviations from the climatology, and thus should be interpreted in a relative sense. A value of zero on the y-axis implies that the events we chose are not

different from the climatology of the days they occurred in.

Concerning the second part of the Reviewer's question, $d$ and $\theta$ relate to different aspects of the atmosphere's intrinsic predictability. The local dimension is a proxy for the "complexity" (here we use the term in a very broad and mathematically imprecise sense) of the evolution of the atmosphere about a given reference state. The persistence tells us how rapidly the evolution described by $d$ happens. While the two metrics are overall correlated, there are cases where they diverge. In such cases, there is no exact rule to determine which of the two will "dominate", and an evaluation must be made in relation to our physical understanding of the weather phenomenon being studied. We will summarize the above points in the revised version of the manuscript.

**Reviewer 2:** L277-283: It is not clear to me what can we learn from the abserr graphs. Is this the error computed relative to the stations?

**Response:** Indeed, this is the error computed relative to the stations. Both the spread and absolute error relate to the practical predictability as defined in Sect. 1 and in Sect. 2.4. We will elaborate on the use of both scores to estimate the practical predictability in Sect. 2.4 of the revised text. Specifically, the correlation between the ensemble spread and skill of the NWP model indicates how well the ensemble describes a priori the practical predictability of the atmospheric configuration we are considering (Whitaker and Loughe, 1998).

**Reference:**

Whitaker JS, Loughe AF. 1998. The relationship between ensemble spread and ensemble mean skill. *Monthly Weather Review* **126**: 3292–3302. https://doi.org/10.1175/1520-0493(1998)126<3292:TRBESA>2.0.CO;2

**Reviewer 2:** L306: What do we learn from Fig. 8? Please enhance its description or cut it out of the main body (it could be transferred to the supplementary).

**Response:** We plan on extending the description of Figure 8, which describes the large-scale situation during the 2010 heat wave. We will especially focus on the effect the Rossby wave breaking may have on the generation of this extreme heat wave.

**Reviewer 2:** L390: Could you also provide a table showing the d and Theta for the analyzed times?

**Response:** We will add the computed *d* and *θ* metrics to the KIT open data repository (https://www.bibliothek.kit.edu/english/kitopen.php).

**Reviewer 2:** Figure 1: The color scale of panels a and b is not the same (panel a uses green colors in the middle of the SLP range, while panel b uses only yellows). Please match the color scales. Furthermore, please either write the interval of the SLP contours or add labels to some of them, so it could be easier to compare between the plots.

**Response:** We will revise the Figure according to the Reviewer's suggestion (see new Figure 1 above).

**Reviewer 2:** Figure 3: either the SLP and Z500 labels were swapped or their mentioning in the figure caption. Please also explain what is represented by each dot. Is it the 12 UTC d and theta from the NCEP for each of the analyzed days? If so, please write something in that spirit.

**Response:** We do not believe the labels were swapped, but we will recheck this and fix the labels accordingly. Regarding the second question from the Reviewer, the dots represent the daily values of *d* and *θ* for days either exceeding the 90% of CSI or below the 10% of CSI. We will add this in the caption of Figure 3.

**Reviewer 2:** Figure 7a, b: Could you please make the blue colors somewhat transparent? It is harder to read the map in the opaque form of the trajectory's densities.

**Response:** We will make the colors transparent so the Figure is more readable. Please see the new version of Figure 7a, b below.

[Figure]

**New Figure 7a, b** Same as Figure 7a, b, but with transparent blue colors.

**Reviewer 2:** Figure S1: Please add either topography or some measure of the summer-climate (average temp. / max temp.) to the map. In this way the readers could assess why are the 5 stations are representative of the climate.

**Response:** We will add the average summer (July-August) temperature to the Figure, in order to better show the representative nature of the stations. Please see new Figure S1 below.

[Figure]

**New Figure S1** The five homogenized stations on top of the average summer (July-August) temperature over Israel for 1995-2009 (https://ims.gov.il/he/climateAtlas; shading in color - ˚C).

**Reviewer 2:** Technical corrections L46: "On the upper levels": please consider adding the words "of the troposphere".

**Response:** The words will be added according to the Reviewer's suggestion.

**Reviewer 2:** L49: "Saaroni and Ziv,2000": please add a space before the "2000".

**Response:** A space will be added.

**Reviewer 2:** L89: Please consider deleting the word "thus".

**Response:** The word 'thus' will be deleted in the revised version of the manuscript, according to the Reviewer's suggestion.

**Reviewer 2:** L141: ". . . nine out of eleven days": please add "on average".

**Response:** The words will be added in the revised version of the manuscript, according to the Reviewer's suggestion.

**Reviewer 2:** References: Kushnir, Y., Dayan, U., Ziv, B., Morin, E. and Enzel, Y.: Climate of the Levant: phenomena and mechanisms, in Quaternary of the Levant: environments, climate change, and humans, edited by Y. Enzel and B.-Y. Ofer, pp. 31–44, Cambridge University Press, Cambridge, UK., 2017.

**Response:** Thank you for providing this reference. It will be cited in the revised version of the manuscript, according to the Reviewer's suggestion.

---

## Author Response (AR1)

Institute for Meteorology and Climate Research Department of Tropospheric Research

KIT-Campus Nord | IMK-TRO | 76344 Eggenstein-Leopoldshafen

Prof. Sagnik Dey Editor Earth System Dynamics Dr. Assaf Hochman

Hermann-von-Helmholtz-Platz 1 76344 Eggenstein-Leopoldshafen Telefon: 0721-608-2-4942 Fax: 0721-608-2-4742 E-Mail: assaf.hochman@kit.edu Web: http://www.imk-tro.kit.edu/ Datum: 1.12.2020

**Subject: Submission of our revised manuscript on the predictability of heat waves over the Eastern Mediterranean**

Dear Prof. Dey,

Enclosed please find the revised manuscript entitled: "A New View of Heat Wave Dynamics and Predictability over the Eastern Mediterranean", by Assaf Hochman, Sebastian Scher, Julian Quinting, Joaquim G. Pinto and Gabriele Messori.

We have addressed all of the concerns of Reviewer 1 and 2 according to the point by point response.

We would like to thank the editor and anonymous Reviewers for their valuable time and useful contributions, which definitely helped to improve our manuscript. We look forward to receiving feedback on the updated manuscript and would be glad to respond to any further questions and comments.

Sincerely yours,

Dr. Assaf Hochman

Karlsruher Institut für Technologie (KIT) Kaiserstraße 12 76131 Karlsruhe USt-IdNr. DE266749428 Präsident: Prof. Dr.-Ing. Holger Hanselka Vizepräsidenten: Prof. Dr. Thomas Hirth, Prof. Dr. Oliver Kraft, Christine von Vangerow, Prof. Dr. Alexander Wanner LBBW/BW Bank IBAN: DE44 6005 0101 7495 5001 49 BIC/SWIFT: SOLADEST600 LBBW/BW Bank IBAN: DE18 6005 0101 7495 5012 96 BIC/SWIFT: SOLADEST600 esd-2020-37: "A new view of heat wave dynamics and predictability over the Eastern Mediterranean" by Assaf Hochman, Sebastian Scher, Julian Quinting, Joaquim Pinto and Gabriele Messori.

**Point by point response to Reviewer 1:**

**Reviewer 1:** In this paper, the authors employ an approach from dynamical systems theory to quantify the (intrinsic) predictability of atmospheric states based on reanalysis data during cold and hot extremes over the Eastern Mediterranean. This is complemented with GEFS reforecasts, which are used to infer forecast uncertainty, or practical predictability. While the distinction and investigation of practical and intrinsic predictability is not new (e.g. Melhauser & Zhang, 2012), I am not aware of any comparable publications in the context of heatwaves. In addition, a simple Lagrangian model is used to reveal the origin of near-surface air masses during hot and cold extreme events. The resulting paper is nicely structured, not too lengthy and certainly an interesting read. I only have two minor comments and a few additional comments, questions and suggestions, as the manuscript is well written and understandable. **Response:** Thank you for the positive feedback. We have addressed all of the Reviewer's comments in the revised version of the manuscript as described below.

**Reviewer 1:** In Melhauser & Zhang (2012), the classic Lorenz (1969) paper is cited multiple times; whereas intrinsic predictability is first defined as "the extent to which prediction is possible if an optimum procedure is used", it is then also related to knowledge of the 'atmospheric state'.

Now, here, the authors do not cite any study when they claim that "As opposed to the practical predictability, the intrinsic predictability only depends on the characteristics of the atmosphere itself." Even though I understand that authors might want to stick to historical definitions, to me, it makes only little sense to limit the forecasting system to the atmosphere. An increasing amount of evidence shows that the Earth's surface does not only supply the atmosphere with heat and moisture, but, to some extent, also exerts control over it (e.g. Koster *et al.*, 2010; Dirmeyer *et al.*, 2018). Knowledge of the land (and ocean) surface state thus implies improved predictability up to sub-seasonal timescales (thanks to, e.g., soil memory).

To be clear, I believe the focus on the atmosphere in this study makes sense, but I would still like to question why this 'intrinsic' predictability should be purely atmospheric by any means. To me, it seems like Lorenz (1969) emphasized the knowledge of all governing equations as well as observing the initial state, and I do not see why this would not include

other components of the Earth System.

**Response:** We agree with the Reviewer's comment. Indeed, the atmospheric state depends not only on the atmosphere, but is influenced by interactions with land and ocean. However, it is important to note that in many – albeit certainly not all – cases these interactions influence the atmosphere at time scales longer than those we consider in our analysis (e.g., Entin *et al.*, 2000), and act as a seasonal-scale preconditioning to extremely high summer temperatures (e.g. the mechanism discussed in the Zampieri *et al.* (2009) study the Reviewer cites in his second comment). We have rephrased the sentences related to this notion. We have specifically clarified that, while we take a predominantly atmospheric perspective, centered on synoptic timescales, there is ample evidence of the importance of surface interactions with other "spheres" of the climate system, not least for controlling atmospheric predictability (such as in the experiments described in the Koster *et al.* (2010) study the Reviewer pointed us to). In this respect, we read and referenced the relevant literature suggested by the Reviewer. The added text can be found in lines: 15-17, 84, 90-92, 419-422 of the revised version of the manuscript.

**Reference:**

Entin JK, Robock A, Vinnikov KY, Hollinger SE, Liu S, Namkhai A. 2000. Temporal and spatial scales of observed soil moisture variations in the extra tropics. *Journal of Geophysical Research* 105(D9): 11865–11877. https://doi.org/10.1029/2000JD900051

**Reviewer 1:** Concerning the 2010 heatwave analysis: while it is interesting to show that the air parcels tracked back in time were warmer than on average even 10 days in the past in this specific case, Bieli *et al.* (2015), e.g., already found that high temperatures in the Balkans area tend to be primarily the result of high starting temperatures combined with extensive descent, enabling strong adiabatic heating. The authors also attempt to explain why particularly the metrics calculated on SLP differ so strongly from those calculated on Z500. I am using this opportunity to refer to my previous comment here – perhaps the fact that the evolution of the atmospheric state closer to the surface tends to follow less of a clear pattern, or "the larger spread in dynamical systems properties across the different heat waves for SLP than for Z500", is partially caused by interactions with the land surface. Naturally, these interactions predominantly affect the lowermost parts of the troposphere, and to provide an example, it can actually be seen (if the different units are accounted for) in Fig. 12 of Zampieri *et al.* (2009) that unusually dry soils affect SLP more than Z500 in a modelling experiment. I thus think that

the 2010 heatwave part needs a bit more attention, as currently, the main message is that the single case is similar to the climatology with respect to Z500 evolution during heatwaves, but highly different in terms of SLP, and in my opinion, this is not explained sufficiently (see also comments above and concerning L. 296 below).

**Response:** Thank you for this comment and for pointing us to some useful references we had overlooked. We agree that the discussion of the 2010 heat wave could be extended with respect to the differences between the dynamical systems analysis close to the surface and at 500 hPa. We now discuss the possible influence air-sea interactions may have on the dynamical systems metrics at surface level in the revised manuscript. Soil moisture, on the other hand, may not be an important factor for controlling heat waves over the South-Eastern part of the Mediterranean, but rather more important in the North-Eastern areas (Zittis *et al.*, 2014). It may however be possible that low soil moisture in the regions where the air parcels originate influence the intensity of Eastern Mediterranean heat waves. We have added a discussion of this point in the revised manuscript. We have further read the references suggested by the Reviewer and we referred to them in the revised version of the manuscript were applicable. The revised text can be found in lines: 361-364, 386, 417-422 of the revised version of the manuscript.

**Reference:**

Zittis G, Hadjinicolaou P, Lelieveld J. 2014. Role of soil moisture in the amplification of climate warming in the eastern Mediterranean and the Middle East. *Climate Research* 59: 27-37. https://doi.org/10.3354/cr01205

Reviewer 1: L. 36: Feeling the heat, 2018 (commanissing?)

**Response:** A comma has been added. This can be found in line 37 of the revised version of the manuscript.

**Reviewer 1:** L. 49: Saaroni and Ziv,2000 (spacemissing?)

**Response:** A space has been added. This can be found in line 55 of the revised version of the manuscript.

**Reviewer 1:** L. 126: If the CSI comprises the boundary layer height, then why is it absent from the equation below (L. 130)? Also, in the cited Saaroni *et al.* (2017), the atmospheric boundary layer height itself is barely mentioned, but rather the (height of the) persistent marine inversion. I thus recommend slightly editing (or shortening) this part to further enhance the

consistency and clarity of the text.

**Response:** We agree with the Reviewer. We have reformulated this part of the manuscript according to Saaroni et al. (2017). The revised text can be found in lines: 59 and 135 of the revised version of the manuscript.

**Reviewer 1:** L. 212: Is there any reason for this choice, i.e. initializing trajectories between the surface and 90 hPa above, other than simplicity? To me, it seems more intuitive to always track the air masses within the atmospheric boundary layer back in time, whose height may vary from day to day, and tends to be (positively) anomalous particularly during heatwaves (this might not be the case in the study area of interest, but was certainly true for the 'epicenter' of the 2010 Russian heatwave; see, e.g. Miralles *et al.*, 2014). However, considering that Dayan *et al.* (2002) demonstrated onlylittle synoptic-scale influence on summer ABL heights compared to the distance to the coast, the usage of a constant layer to be tracked backward might be entirely justified, but perhaps the authors can still elaborate on their choice.

**Response:** Thank you for this comment. According to recent studies, the planetary boundary layer height in Israel during summer is  $\sim 600 - 900$ m above the surface (Uzan *et al.*, 2016; 2020). Assuming that the pressure decreases by 1 hPa every 8m height difference, 90 hPa corresponds to about 720m, thus 90hPa can be considered a reasonable choice.

We have further evaluated the trajectories that start from 90hPa above the surface and higher (Figure R1). We find that at these levels the picture is somewhat different. Air parcels associated with heat waves are located over the Mediterranean Sea and east of Israel prior to the heat wave (Figure R1a). Air parcels associated with cool days originate mostly over the Atlantic and Europe and are transported downstream (Figure R1b). This further indicates that our trajectory analysis successfully identifies air parcels in the boundary layer, which have a different path than those in the free troposphere. Moreover, the thermodynamic properties of the air parcels do not show any noticeable differences between heat waves and cool days (Figure R1c-f). We have elaborated on the choices we made with regards to the back trajectories' analysis as described above in the revised version of the manuscript. The revised text can be found in lines: 272-275 of the revised version of the manuscript.

**Reference**

Uzan L, Egert S, Alpert P. 2016. Ceilometer evaluation of the eastern Mediterranean summer boundary layer height – first study of two Israeli sites. *Atmospheric Measurement*  Techniques 9: 4387–4398. https://doi.org/10.5194/amt-9-4387-2016

Uzan L, Egert S, Khain P, Levi Y, Vadislavsky E, Alpert P. 2020. Ceilometers as planetary boundary layer height detectors and a corrective tool for COSMO and IFS models. *Atmospheric Chemistry and Physics* 20: 12177-12192. https://doi.org/10.5194/acp-20-12177-2020